# Recurrent neural networks with explicit representation of dynamic latent variables can mimic behavioral patterns in a physical inference task

Rishi Rajalingham[1], Aída Piccato[1,2] & Mehrdad Jazayeri [1,2] ✉

Primates can richly parse sensory inputs to infer latent information. This ability is hypothesized to rely on establishing mental models of the external world and running mental simulations of those models. However, evidence supporting this hypothesis is limited to behavioral models that do not emulate neural computations. Here, we test this hypothesis by directly comparing the behavior of primates (humans and monkeys) in a ball interception task to that of a large set of recurrent neural network (RNN) models with or without the capacity to dynamically track the underlying latent variables. Humans and monkeys exhibit similar behavioral patterns. This primate behavioral pattern is best captured by RNNs endowed with dynamic inference, consistent with the hypothesis that the primate brain uses dynamic inferences to support flexible physical predictions. Moreover, our work highlights a general strategy for using model neural systems to test computational hypotheses of higher brain function.

From just a few glances, we can parse the structure of a novel scene, generate a rich understanding of its components, and use this understanding to make general inferences and predictions[1]. This understanding helps us infer the latent states of objects and events, predict plausible and implausible future states, plan intervening actions, and anticipate the consequences of those actions. Despite the centrality of these capacities in human intelligence, the underlying computations remain unknown.

A dominant theory is that the brain constructs mental models of the physical world and relies on mental simulations of those models for making inferences[1–3]. Mental simulation is broadly defined as the capacity to imagine "what will or what could be"[4], and is also thought to underlie other cognitive functions such as imagination[5,6] and counterfactual reasoning[7]. Concretely, the mental simulation hypothesis predicts that the nervous system makes inferences in the absence of sensory input by forming a dynamic inference engine that can internally track latent environmental states (see Fig. 1A).

Currently, the strongest evidence in support of this hypothesis comes from comparing human behavior to predictions made by specific dynamic inference engines, such as high-level computer programs that are analogous to how engineers run simulations of physical systems[1,8]. However, many behaviors that are proposed to rely on such dynamic inference engines can also be produced by automatized nonlinear functions such as those implemented by feedforward neural network models[9] (but see ref. 10). Therefore, it is still unclear whether or not the neural systems that support human inference rely on dynamic inference strategies.

Recent advances in artificial neural networks have created new opportunities to go beyond abstracted process models, and instantiate and rigorously test specific hypotheses about how neural systems solve various perceptual, cognitive, and motor tasks[11–16]. Here, we adopted a similar approach combining suitable task design, human and monkey (hereafter, "primate") behavior, and artificial neural network modeling to test the dynamic inference engine hypothesis.

[1]McGovern Institute for Brain Research, Massachusetts Institute of Technology, Building 46, 43 Vassar St., Cambridge, MA 02139, USA. [2]Department of Brain & Cognitive Sciences, Massachusetts Institute of Technology, Building 46, 43 Vassar St., Cambridge, MA 02139-4307, USA. ✉e-mail: mjaz@mit.edu

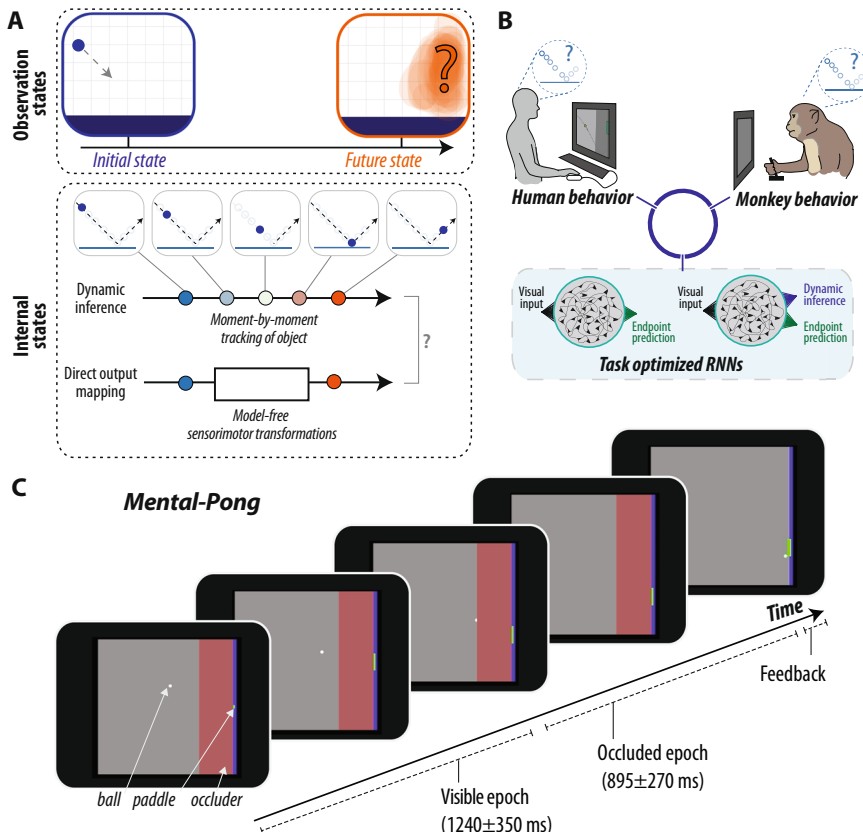

**Fig. 1 | Hypotheses and experimental strategy. A** Physical predictions. A dominant theory in cognitive science is that humans make inferences about physical processes using "mental simulations" of the physical world. To illustrate this, the top panel depicts observations of the external world, a ball falling towards the ground. In making an inference about the future state of this world, the brain could form a dynamic inference engine, i.e., dynamically track latent environmental states to generate behavioral outputs. Alternatively, the brain could support such inferences via automated nonlinear function approximations, without explicitly tracking latent environmental states. **B** Experimental strategy. We tested the mental simulation hypothesis by directly comparing the behavior of humans, monkeys,

and task-optimized recurrent neural network models. We trained RNN models to solve the same task as humans and monkeys (see panel **C**), but additionally optimized some models to perform dynamic inference of the latent position of the ball. **C** Behavioral task. We developed a behavioral task (*M-Pong*, reminiscent of the computer game called Pong) that aims to probe the ability of humans and monkeys to flexibly and rapidly predict the future state of a previously learned rich physical world. The objective of the task is to control the vertical position of a paddle to intercept a ball moving across a two-dimensional frame with reflecting walls and an occluder (see Methods for details). M-Pong requires making inferences about the dynamics of an unobserved external process.

Specifically, we compared the behavior of primates in a ball interception task with a partially occluded ball to artificial neural network models with or without dynamic inference abilities. We focused on a relatively simple ball interception task with a few considerations in mind. First, since primates have a natural ability to intercept moving objects, we hypothesized that they would easily learn this task and that their behavioral characteristics would reveal their inductive biases, and not specializations that may result from overtraining. Second, the task is particularly suitable for testing the dynamic inference hypothesis because, in principle, it can be solved both by a dynamic inference engine (i.e., by performing a moment-by-moment tracking of the latent ball) and an automated nonlinear function (e.g., by estimating the final ball position using a static nonlinear geometric computation from only the initial ball state, which does not require moment-by-moment tracking). In other words, dynamic inference is not an obligatory consequence of task design but rather a solution that primates might plausibly adopt. Finally, we included monkeys in our experiment with an eye toward future neurophysiology experiments to investigate the underlying neural mechanisms and validate the aforementioned hypotheses.

To test the dynamic inference hypothesis, we here compare the behavior of humans and monkeys in the task to that of a large battery of recurrent neural networks (RNN) with or without dynamic inference capacities (Fig. 1B). We find that the behavioral similarity of the RNNs

to primates is specifically and systematically related to the degree to which RNNs faithfully track latent environmental states. Further analysis reveals how structured dynamic representations in the primate-like RNNs serve as a computational substrate for dynamic inferences. Taken together, these results are consistent with the hypothesis that the primate brain uses dynamic inferences to support physical predictions, and establish a platform for uncovering the neural mechanisms underlying primate physical inferences.

## Results

### Behavioral task

We devised a behavioral task in which participants had to control the vertical position of a paddle to intercept a ball moving across a two-dimensional frame with reflecting walls (Fig. 1C). The ball's initial position and velocity (speed and heading) were randomly sampled on every trial (Fig. S1). Moreover, the frame contained a large rectangular occluder before the interception point such that the ball's trajectory was visible only during the first portion of the trial. Depending on the ball's initial position and velocity, the visible and occluded portions of each trial lasted 1240+/−350 and 895+/−200 ms, respectively. Participants initiated each trial by fixating on a central fixation dot, but were subsequently free to make any eye movements. On every trial, participants could move the paddle as soon as the ball began to move, and could drive it freely and at a constant speed in up or down directions

from its initial position at the middle of the screen with the goal of intercepting the ball when it reached the paddle. We used identical task parameters and contingencies for humans and monkeys with two exceptions. First, humans moved the paddle using a keyboard, whereas monkeys used a one-degree-of-freedom joystick. Second, while both monkeys and humans received visual feedback at the end of each trial, monkeys additionally received a small juice reward when they successfully intercepted the ball. We refer to this task as M-Pong because of its similarity to the computer game Pong, and because of the presence of the occluder that necessitates mental (as opposed to visual) computations.

Critically, M-Pong requires that participants make a response in the absence of external stimuli, and furthermore that the appropriate response is a time-varying nonlinear function of previously observed stimuli, in contrast to simple memory tasks that only require the maintenance of previously observed stimuli. Instead, the M-Pong task enforces participants to make inferences that reflect the dynamics of an unobserved external causal process. We note that a plausible strategy to solve the task is to "mentally simulate" the physical movement of the ball as it moves behind the occluder. However, this strategy is not necessary; since the initial position and velocity of the ball fully determine all its future states, a sufficiently nonlinear function would be able to solve the task without any dynamic tracking.

## Comparing human and monkey behavior

We collected behavioral data from 12 humans performing 200 unique task conditions (i.e., different initial ball positions and velocities), all randomly interleaved. For all participants, we measured the position of the paddle throughout the trial (Fig. 2A, left). Participants began moving the paddle early in the trial while the ball was still visible (Fig. S2B) and generated smooth movement trajectories that continued throughout the occluded portion of the trial (Fig. S2B).

We quantified overall performance using the mean absolute error (MAE), computed as the absolute difference between the final ball position and the final paddle position, averaged across all trials and all conditions. Importantly, we went beyond the overall performance metric and additionally analyzed condition-specific errors. For each of the 200 unique conditions, we quantified the average endpoint error, i.e., the signed difference between the final ball position and the final paddle position, averaged across trial repetitions of a specific condition (Fig. 2A, right). This resulted in a 200-dimensional error-vector that we used to compare behavior across humans, monkeys, and RNNs.

The analysis of the error-vector in humans led to two important conclusions. First, the error-vector was highly similar across humans (see Fig. S4A), indicating that humans solve the task using a similar inference strategy. Second, the pattern of errors could not be straightforwardly explained by the initial ball state (i.e., by a linear function of the ball's position and velocity at the start of the trial, and at the start of the occluded epoch; $R^2 < 0.05$ for all, see Fig. S4B). From these observations, we concluded that the common pattern of errors across humans reflects the inference strategy they rely on to solve the task, and can thus serve as a metric to compare with monkeys and RNNs.

Next, we trained two monkeys to perform the M-Pong task. After an initial familiarization with the joystick as the means for controlling the paddle, monkeys were trained to intercept a moving ball without any occluders, which they mastered after several days. Next, we introduced the occluder, interleaving trials where the ball was either completely or partially invisible. To our surprise, monkeys reached high performance under occlusion on the very first behavioral session, and maintained this high performance over subsequent sessions. This is shown in Fig. 2C, which shows that the per-session error of monkeys (green/blue circles) is lower than chance performance (shuffled null, gray dashed line, and arrowhead), and approximately matches the final

performance of humans and monkeys (red/blue arrowheads). Note that the shuffled null does not correspond to the worst possible performance, as it is obtained by shuffling the correct outputs for each condition, and thus preserves the distribution of outputs. Could this performance reflect stimulus-response memorization strategies? To investigate this, we tested monkeys on their ability to generalize to novel M-Pong conditions. While monkeys were trained on the same exact 200 conditions as humans, we used a curriculum that allowed us to test their generalization ability by first introducing 50 of the unique conditions for initial training, and later using the remaining 150 conditions as a held-out test. Both monkeys were able to effortlessly generalize to these test conditions, and maintain the same performance level on the very first trial of the test conditions (Fig. 2C, "test" on abscissa, open circles, $n = 150$ conditions). This generalization performance did not reflect prior exposure to the corresponding visible conditions, which were randomly interleaved on 25% of trials, as evidenced by the similar performance on the subset of test conditions with no such prior exposure (Fig. 2C, "test" on abscissa, closed circles, $n = 112, 106$ conditions for monkey P, C). Taken together, these results demonstrate that monkeys rapidly learn to perform M-Pong and immediately generalize to novel M-Pong conditions, suggesting that the computational demands of M-Pong are broadly compatible with macaque monkeys' inductive biases.

Next, we analyzed monkeys' behavior based on the systematic pattern of errors across the 200 unique conditions. We first verified that error patterns were similar between the two monkeys ($r = 0.74 \pm 0.001$, Fig. S4A), and could not be explained in terms of the ball's initial position and velocity (i.e., by a linear function of the ball's position and velocity at the start of the trial, and at the start of the occluded epoch; $R^2 < 0.05$ for all, see Fig. S4B). Next, we compared the error-vector between monkeys and humans. The conditions that monkeys found to be particularly difficult were similarly difficult for humans (Fig. S3). Moreover, the overall patterns of errors across the entire set of 200 unique conditions were highly similar between humans and monkeys (Fig. 2D). Based on these observations, we concluded that monkeys rely on an inference strategy similar to humans, and that the pattern of errors across primates reflect this common strategy.

To quantify the similarity between monkeys and humans, we developed a summary statistic, which we term human-consistency score. We defined human-consistency score as the degree to which an error-vector was correlated with the error-vector derived from the behavioral responses in humans. To improve our estimate of human-consistency, the correlation coefficients were adjusted for sampling noise (see Methods). Moreover, to avoid overestimating human-consistency, errors were computed as residuals rather than simple differences, equivalent to computing a partial correlation between the paddle endpoint across conditions, accounting for the co-varying pattern of ground truth positions (see Methods). Defined in this way, a human-consistency score of 1 would correspond to identical error patterns and a score of zero, to random errors. Human consistency score for monkeys was large ($0.89 \pm 0.003$) but smaller than the ceiling value estimated based on a comparison of behavior between humans ($0.95 \pm 0.012$, see Methods).

## Comparing primates and recurrent neural network models

RNNs have vast computational capacities and can, in principle, be trained to establish arbitrarily complex functions[17]. The RNN solutions one can find for a specific task are not unique, and can vary with various factors, including network architecture, training protocol, and perhaps most importantly, the optimization constraints one imposes on the solution[18]. We exploited this non-uniqueness property to ask whether RNNs of various architectures that are optimized to perform M-Pong would behave more similarly to primates if they were additionally optimized to dynamically track the ball position (Fig. 3A).

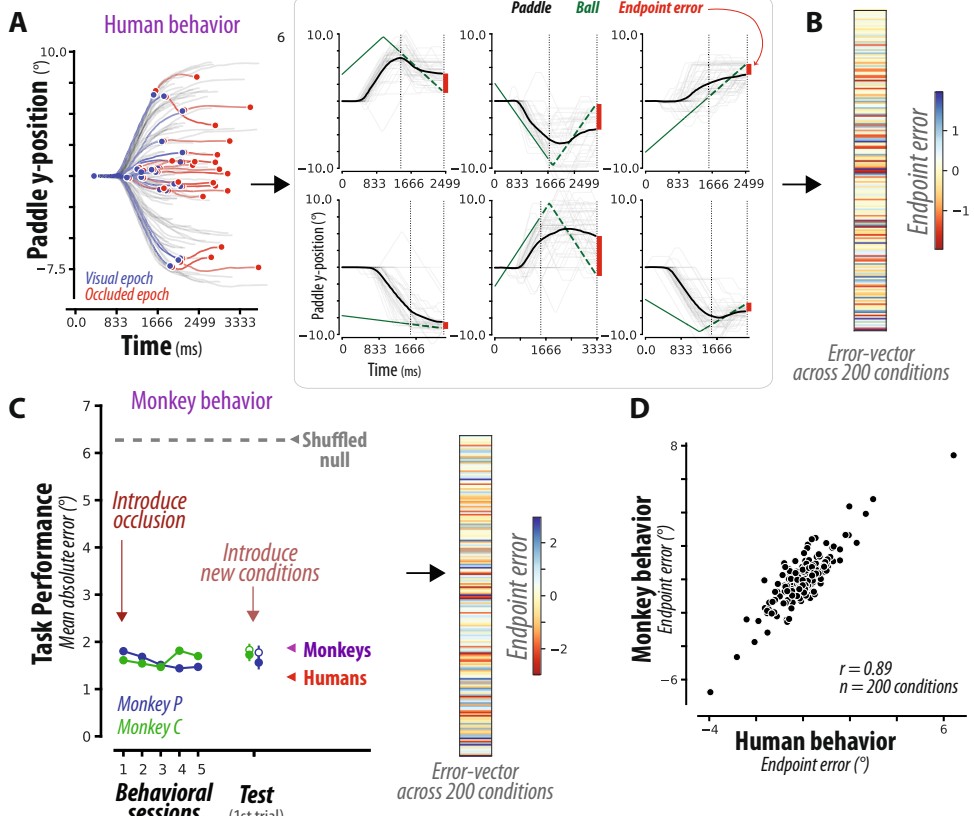

**Fig. 2 | Comparing human and monkey behavior. A** Human behavior. (left) Each curve corresponds to the paddle position (in units of degrees of visual angle) over time for a single task condition, averaged across dozens of trial repetitions, separated into movements while the ball was visible (blue) or occluded (red). (right) For six example task conditions, this average paddle trajectory is overlaid on each individual trial repetition (light gray), as well as the true vertical position of the ball (green), from which the average endpoint error (red) is estimated (see Methods). **B** Across 200 different task conditions, we measure a pattern of endpoint errors (termed error-vector) to characterize behavior. The error-vector is shown as a colored vector, with colors spanning mean ± 2SDs of the error range. **C** Monkey behavior. (left) After training monkeys to manipulate the joystick in order to control a paddle and intercept moving balls, we introduced the occluder, interleaving trials where the ball was completely versus partially occluded. Monkeys reached high performance under occlusion on the very first behavioral session, and maintained this high performance over subsequent sessions. Furthermore, monkeys

maintained high performance when tested on 150 novel task conditions, and this generalization was immediate, with comparable high performance on the very first trial of a new condition for both monkeys ("test" on abscissa, open circles, mean + SE across 150 conditions). This generalization performance did not reflect prior exposure to the corresponding visible conditions, which were randomly interleaved on 25% of trials, as evidenced by the similar performance on the subset of test conditions with no such prior exposure ("test" on abscissa, closed circles, $n = 112, 106$ conditions for monkey P, C). Chance performance (shuffled null), and the final performance of humans and monkeys are shown via the gray, red, and blue arrowheads respectively. (right) Following training, we characterized monkey behavior across 200 task conditions with a pattern of endpoint errors; colors span mean ± 2SDs of the error range. **D** The scatter shows the average endpoint error for each of the 200 conditions, for humans versus monkeys. We observe a remarkable similarity in the error-vectors of humans and monkeys.

We built each RNN as an input-output system that receives dynamic sensory information about the ball and uses a linear readout to generate a scalar output to drive the paddle (Fig. 3B). The database of RNNs we considered varied along several dimensions (Fig. 3C) including cell type (LSTMs versus GRUs), number of cells (10 vs. 20), input representation (pixel information only versus pixel information plus direction of motion), and regularization strategy (L1 versus L2 loss term). Consistent with the goal of the M-Pong task, we trained all networks using a standard performance-optimizing cost function to find solutions that minimize the error between the paddle and ball position along the y-axis at the time of interception with no constraint on how the paddle ought to move throughout the trial.

To examine the behavior of networks with dynamic inference capacities, we augmented the cost function in subsets of networks by requiring the output of additional linear readouts to carry an explicit online estimate of the ball's x and y position (Fig. 3B, C). For each network architecture, we included four optimization strategies: (1) RNNs that were only optimized for the final paddle position without dynamic inference ("no_sim"), (2) RNNs that were additionally

optimized to dynamically infer the ball position when the ball was visible ("vis_sim"), (3) RNNs that were optimized to dynamically infer the ball position throughout the trial ("all_sim"), and (4) RNNs that used different readout channels to dynamically infer the ball position for the visible and occluded portions ("all_sim2"). Given this potential for overlap between these different optimization types, we did not focus our analyses on group comparisons, but sought to characterize RNNs on an equal footing irrespective of the underlying optimization type (see below). Critically, none of our optimization factors were designed to make RNNs reproduce primate behavior. Instead, our goal was to test if any of the additional cost functions associated with dynamic inference ability would enable RNNs of various architectures to spontaneously behave more similarly to primates. All RNNs were trained using up to 212480 unique conditions, and tested on the same held-out set of 200 unique conditions that were tested in humans and monkeys.

We first verified that RNNs with different architectures and optimization choices were able to learn the task, and some were able to achieve performance levels comparable to humans and monkeys

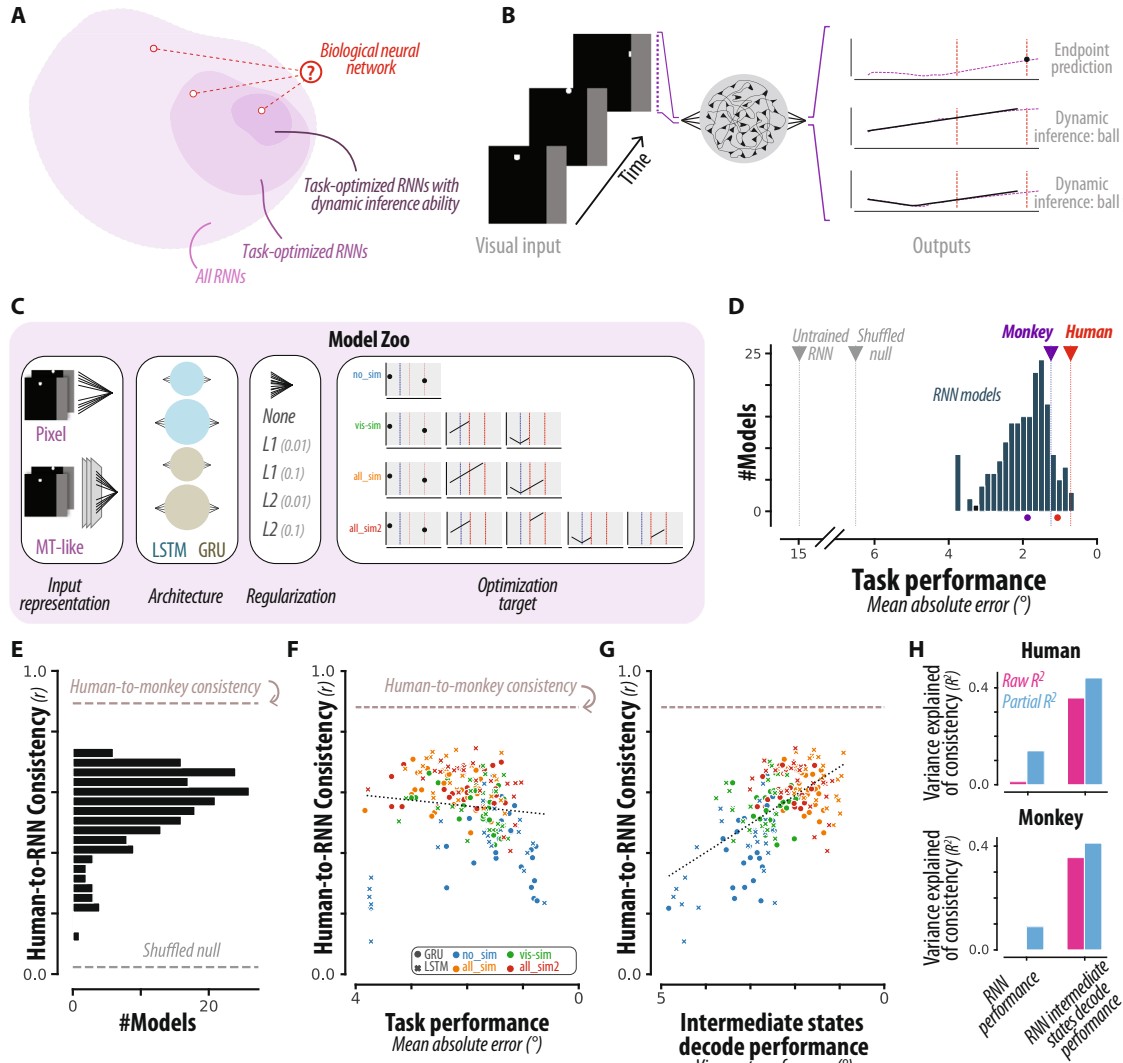

**Fig. 3 | Comparing primates and recurrent neural network models.**
**A** Conceptual schematic. If primate behavior in this task relies on dynamic infer-
ences, then including specific functional constraints (e.g., dynamic inference abil-
ity) on RNNs should lead to finding more brain-like algorithms. **B** RNN behavior. We
trained several hundred RNNs to map a series of visual inputs (pixel frames) to an
endpoint ball y-position (black dot). Some RNNs were additionally optimized to
dynamically track the (x,y) position of the ball throughout the trial (black curves).
The dashed vertical lines on the output panels correspond to the time of occlusion
and the time of the end of the trial. **C** Full set of RNN hyperparameter choices.
Different RNN models varied with respect to architectural parameters and were
differently optimized (either with or without dynamic inference ability). Critically,
RNNs were not optimized to reproduce primate behavior, only to solve the task.
**D** RNN performance. Distribution of RNN performance on this task. Human/mon-
key performance is shown both with (colored circles) and without (colored trian-
gles) the inclusion of errors stemming from trial-by-trial variability. Note that the

abscissa is flipped such that left-to-right corresponds to increasing performance
(i.e., decreasing error). **E** Comparing human and RNN behavior. Distribution of
human-consistency values over all RNNs. Dashed line corresponds to human-to-
monkey consistency (dashed line). **F, G** Functional correlates of human-
consistency. Across all RNNs, scatter of human-consistency versus task perfor-
mance (**F**) and intermediate state decode performance (ISDP, **G**). The variation in
human-consistency across different RNNs did not strongly depend on overall task
performance, but was strongly correlated with ISDP. Note that the abscissas are
flipped such that left-to-right corresponds to increasing performance (i.e.,
decreasing error) and increasing dynamic inference ability (i.e., decreasing ISDP
error). **H** Quantification of functional correlates. The strength of dependence
between the tested functional attributes and consistency with human behavior
(top) and monkey behavior (bottom) is shown as a proportion of variance
explained ($R^2$, pink bars). Partial $R^2$ (blue bars) measures this strength after
accounting for covariations due to the other attribute.

(Fig. 3D). Next, we analyzed the entire zoo of RNNs in terms of the
similarity of their behavior to that of the primates. To quantify the
degree of similarity, we used the condition-dependent average-
endpoint-error that we previously used to compare humans to mon-
keys. Human-consistency scores across RNNs were distributed broadly
and were generally below the values we found for monkeys (Fig. 3E, see
Fig. S5A for group comparisons).

We exploited the variance across RNNs to ask what factors make
certain RNNs behave more or less like primates. As a first step, we
asked whether human-consistency of RNNs could be explained by their
overall performance, which is a common observation in network

models of vision and audition[11,12]. Results revealed no significant rela-
tionship between human-consistency and performance ($R^2 = 0.01$ and
0.00002, $p = 0.104$ and 0.475, for consistency to humans and mon-
keys, respectively, Fig. 3F). This result indicates that the overall per-
formance is not a good metric for identifying the factors that make
RNNs more or less similar to humans.

Next, we focused on our primary objective of using our large
RNN model zoo to test the role of dynamic inference in M-Pong.
Specifically, we asked whether networks that were optimized for
simulating the ball position had higher human-consistency scores.
To compare all RNNs on the same footing, we developed a metric to

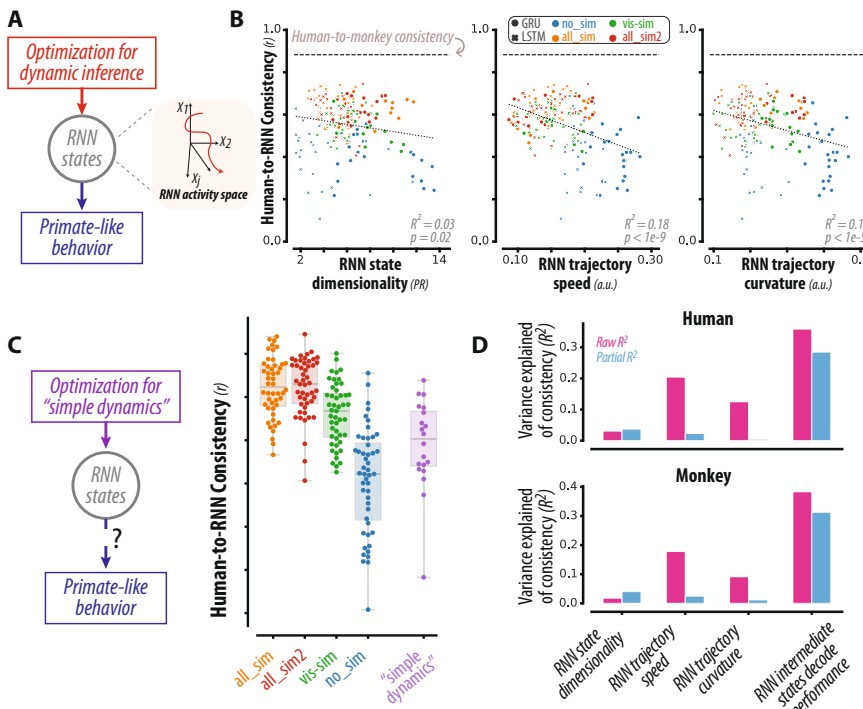

**Fig. 4 | RNN Dynamics underlying primate-like behavior. A** Optimization for dynamic inference ability influences the network behavior by acting via the intermediary of its internal representations. We characterized the structure of the internal representations of RNNs via specific geometric metrics. **B** Each panel shows the scatter over all RNN models of a representational attribute (dimensionality, speed, and curvature) against human-consistency. RNNs with simple dynamics (i.e., slow, smooth and low dimensional representations) exhibited more human-like behavior ($p < 0.02$, 0.001, and 0.001 for dimensionality, speed, and curvature respectively). **C** (left) To assess whether simple dynamics are sufficient to capture primate-like behavior, we constructed new RNN models that were identical to the dynamic-inference-optimized models, except for optimization consisting of task performance and specific regularization terms to favor slow and smooth dynamics. (right) Distribution of human-consistency scores for all RNNs, grouped by optimization types; the swarm plot shows individual models, and the boxplot shows the median, 1st and 3rd quartiles, and range of each distribution. RNNs optimized for simple dynamics better matched human behavior than RNNs optimized for task performance alone (blue vs. purple distributions, $p < 0.001$, $n = 48$, 40 in each group), but failed to capture primate behavior as well as models optimized for dynamic inference (orange vs. purple distributions, $p < 0.001$, $n = 48$, 40 in each group). **D** Across all RNN models, the strength of dependence between the tested representational attributes and consistency with human behavior (top) and monkey behavior (bottom) is shown as a proportion of variance explained ($R^2$, pink bars). Partial $R^2$ (blue bars) measures this strength after accounting for covariations due to all other attributes. We found that dynamic inference ability (quantified via the ISDP) better-predicted consistency scores than all other attributes with respect to both human and monkey behavior.

quantify the degree to which a network carries explicit information about the instantaneous position of the ball behind the occluder, and that could be computed in all networks. This metric, the intermediate state decode performance (ISDP), was computed as the mean absolute error between the true time-varying ball position and the predicted position from a cross-validated linear decoder (see Methods). Results revealed a strong relationship between ISDP and human consistency scores across the RNNs (Fig. 3G). The ISDP was able to explain a large portion of the variance across humans and monkeys independently ($R^2 = 0.36$ and 0.39, $p < 0.001$ and 0.001, for consistency to humans and monkeys, respectively), and the effect was even stronger when we accounted for covariations due to performance ($R^2 = 0.44$, 0.43, for consistency to humans, monkeys respectively Fig. 3H), and greater than the corresponding estimates for task performance ($p < 0.001$, 0.001). We found this to be true even when characterizing behavior using alternative metrics based on eye position (Fig. S8, using error with respect to eye position at various time points during the occluded epoch) and the paddle position (Fig. S9, the paddle error relative to the visible condition endpoint paddle position, and using the time course of paddle-to-ball positions). In sum, RNNs that carried explicit (linearly decodable) information about the latent position of the ball behind the occluder (i.e., performed dynamic inference) were able to capture primate behavioral patterns more accurately than those that did not.

## Dynamics underlying primate-like behavior

How might optimization for dynamic inference lead to networks with more primate-like behavior? Given that this optimization influences the network's output behavior by acting via the intermediary of its internal representations (Fig. 4A), and in light of prior work suggesting that RNNs with slow and smooth internal dynamics tend to best capture neural activity in the primate brain, we here asked if similar dynamic properties of RNNs' internal representations contributed to emulating primate-like behavior. To characterize RNN dynamics, we focused on dimensionality, slowness, and geometric curvature—three attributes that can be quantified in biological networks and can thus serve as predictions for future experiments on the primate brain. We estimated the dimensionality of RNN activity dynamics throughout the entire trial, spanning both visible and occluded epochs, via a participation ratio metric, a weighted sum of the eigen-spectrum obtained from PCA; we estimated the slowness and curvature metrics based on normalized estimates of the first and second temporal derivatives of the same activity dynamics (see Methods for details).

Interestingly, we found that all three attributes were related to human-consistency scores across RNNs. Networks that exhibited "simple dynamics"—i.e., whose activity representations were lower dimensional, lower speed, and lower curvature – better predicted behavioral patterns of humans (Fig. 4B). We quantified the relationship between these attributes and consistency scores by measuring the proportion of variance ($R^2$) each feature could explain about

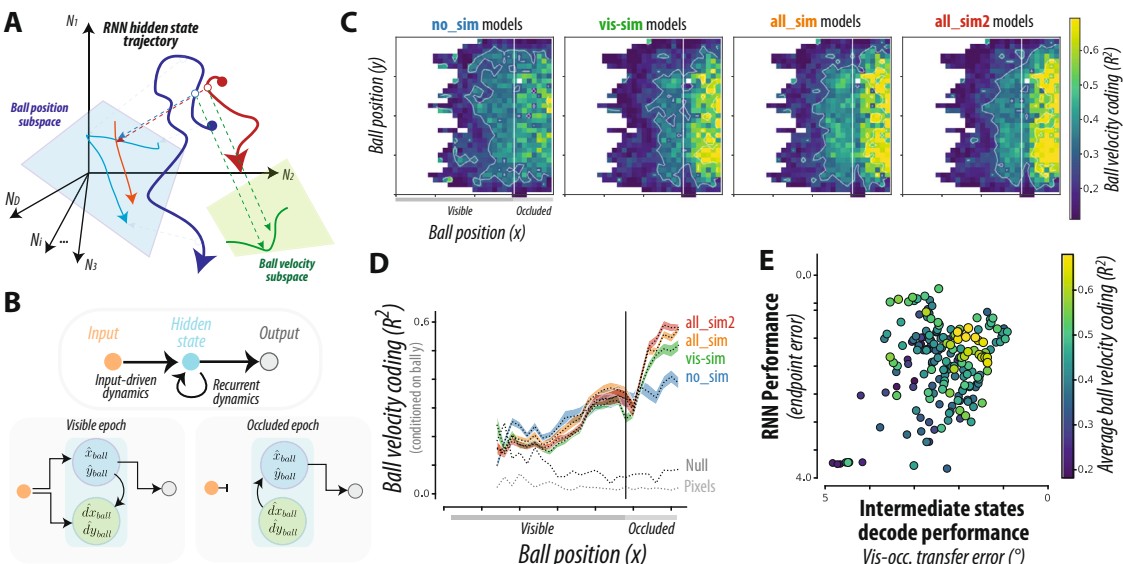

**Fig. 5 | Dynamics underlying computations performed by RNNs. A** Conceptual schematic. Schematic of RNN hidden state trajectory for two hypothetical conditions (dark blue, dark red) during the occluded epoch. To accurately perform dynamic inference, a projection (illustrated in cyan) of each RNN hidden state trajectory must approximately correspond to the latent time-varying position of the ball [x,y] (here illustrated in light blue, light red). However, during the occluded epoch, RNNs are *autonomous* nonlinear dynamical systems, wherein the hidden states evolve over time based on recurrent computations alone. Thus, we reasoned that RNN states additionally encode the ball velocity (green projection). **B** To produce different hidden state trajectories for different conditions, the RNN hidden state trajectories must be flexibly controlled by their initial conditions. Thus, we reasoned that RNNs must estimate both ball position and velocity from the sensory input during the visible epoch (left) and maintain ball velocity estimate throughout the occluded epoch to autonomously update the ball position (right). **C** Each panel shows a heat map of the M-Pong frame, with colors corresponding to the cross-validated proportion of variance of ball velocity explained ($R^2$) by a linear read-out of the RNN hidden states, conditioned on ball position, and averaged over all RNNs of the same optimization type. **D** Ball velocity coding conditioned on the vertical ball position, averaged over all RNNs of the same optimization type. The dashed lines correspond to controls, verifying that velocity coding was not present in the sensory inputs ("pixels"), and not confounded with ball position ("null"). Shaded lines correspond to mean ± SE over RNN models. **E** Over all tested RNN models, the scatter shows the overall performance versus the ISDP index, colored by the strength of the explicit representation of ball velocity during the occluded epoch. Ball velocity predictivity was greater for specific high-performing dynamic inference-based RNN models, with strong correlations to both overall performance and ISDP. Note that the abscissa and ordinate are flipped such that left-to-right and bottom-to-top correspond with increasing performance (i.e., decreasing error) and increasing dynamic inference ability (i.e., decreasing intermediate state decode error).

---

consistency scores. Together, the three attributes predicted a modest proportion of variance associated with consistency scores ($R^2 = 0.18$, and 0.16, $p < 0.001$ and 0.001; with respect to human and monkey respectively). In other words, the state dynamics were generally smoother for RNNs that exhibited more primate-like behavior.

This observation raises the possibility that "simple dynamics" is sufficient for RNNs to emulate primate-like behavior. To address the possibility, we constructed new RNN models (termed "simple_dynamics") that were identical to the dynamic-inference-optimized models in all respects except for their optimization, which consisted of both task performance and specific regularization terms to favor slow and smooth dynamics (see Fig. 4C left, Fig. S7A, Methods). As shown in Fig. 4C (right), the consistency scores of these models were significantly improved relative to models optimized for task performance alone ($p < 0.001$, 0.001; two-tailed Wilcoxon-Mann-Whitney test comparing "no_sim" vs. "simple_dynamics" models for consistency to human and monkey behavior, respectively). However, they failed to capture primate behavior as well as models optimized for dynamic inference (Fig. 4C; $p < 0.001$, 0.001; two-tailed Wilcoxon-Mann-Whitney test comparing "all_sim2" vs. "simple_dynamics" models for consistency to human and monkey behavior, respectively).

To quantify the relative importance of "simple dynamics" versus "dynamic inference ability" on consistency scores, we measured the proportion of variance ($R^2$) of consistency scores that could be explained by each attribute, over all RNN models, including those optimized for simple_dynamics. We found that dynamic inference ability (quantified via the dynamic inference index) best-predicted consistency scores with respect to both human and monkey behavior

(Fig. 4D, pink bars; $R^2 = 0.35$, 0.38, $p < 0.001$, 0.001, for consistency with respect to human and monkey behavior, respectively). This effect could not be explained by covariations due to simple dynamics, as evidenced by significant partial $R^2$—the proportion of variance in consistency scores that could be uniquely explained by dynamic inference ability (Fig. 4D, blue bars; $R^2 = 0.28$, 0.31 for consistency with respect to human and monkey behavior respectively; $p < 0.005$ for comparisons to all three dynamics attributes).

Taken together, these results suggest that optimization for dynamic inference drives RNNs to learn specific activity representations characterized by both simple dynamics (low dimensionality, speed, and curvature) as well as explicit task representations (i.e., linear projections of RNN states matching the ball position). These representations are characteristic of RNN models that exhibit primate-like behavior, and may be analogous to ones found in the primate brain.

## Dynamics underlying computations performed by RNNs

So far, we have treated RNNs as input-output "black box" instantiations of specific cognitive hypotheses. However, several recent studies[13,14,16,19–23] have shown the utility of reverse-engineering RNNs[24] to shed light on how biological neural networks might perform task-relevant computations[25]. In this vein, we sought to understand how the dynamics of RNNs could support the computations necessary for dynamic inference-based M-Pong performance.

Figure 5A schematically illustrates the RNN hidden state space, with trajectories for two hypothetical trial conditions (dark blue, dark red) during the occluded epoch. We note that in order to accurately perform dynamic inference, a projection of each trajectory must

approximately correspond to the latent time-varying position of the ball [x,y] (light blue, light red, Fig. 5A). During the visual epoch of the task, the position variables may be computed through nonlinear transformation of direct sensory input to the network. In contrast, during the occluded epoch when all sensory inputs have extinguished, RNNs behave as autonomous dynamical systems, and must therefore update the position based on a nonlinear function of network hidden states at the previous time steps. Since updating the position requires information about velocity, we reasoned that RNN states during the occluded epoch should additionally represent the ball velocity [dx,dy] (green subspace, Fig. 5A). Critically, given the nonlinearities present in RNN units, such velocity representation need not be explicit, i.e., accessible via a linear read-out.

An autonomous RNN with no external input can behave differently for different trial conditions if and only if its initial condition at the beginning of the occluded epoch is appropriately adjusted (i.e., different initial conditions for different trial conditions). Accordingly, we reasoned that RNNs must use the visual epoch to establish an appropriate condition-dependent initial condition for the subsequent occluded epoch. This hypothesis is schematically illustrated in Fig. 5B, showing that RNNs estimate both ball position and velocity from the sensory input during the visible epoch (left) and maintain ball velocity estimate throughout the occluded epoch to autonomously update the ball position (right).

These considerations can be summarized in terms of a bipartite hypothesis that RNNs must use the visual information to extract velocity information during the visual epoch, and must have a representation of this velocity information during the occluded epoch. To test this hypothesis, we quantified the extent to which ball velocity is encoded in the hidden state representation of RNNs. Each panel in Fig. 5C shows a heat map of the M-Pong frame, with colors corresponding to the cross-validated proportion of variance of ball velocity explained ($R^2$) by a linear read-out of the RNN hidden states, averaged over all RNNs of the same optimization type (see Fig. S7B-left for each individual RNN model, and Fig. S7B-right for corresponding results using a single linear read-out). We observe the emergence of a representation of ball velocity during the visible segment, consistent with the first part of the hypothesis. Moreover, there was strong and persistent velocity coding at nearly all ball positions throughout the occluded segment, consistent with the second part of the hypothesis. This is further quantified in Fig. 5D, which shows the ball velocity coding, conditioned on the vertical ball position. Moreover, we verified that such ball velocity coding was not present in the sensory inputs ("pixels" in Fig. 5D), and not confounded with ball position ("null" in Fig. 5D). Interestingly, the velocity coding was not simply maintained, but increased during the occluded epoch, suggesting that the representation of ball velocity is more linearly decodable in the recurrent dynamics than in the input-driven dynamics (where nonlinear computations from the pixel inputs are expected).

Interestingly, Fig. 5C, D suggest an enhanced representation of ball velocity in RNN models that were directly optimized for dynamic inference (e.g., all_sim, all_sim2). Indeed, across RNNs, explicit ball velocity coding during occlusion was correlated with both overall performance ($r = 0.57$, $p < 0.001$) and intermediate state decode performance ($r = 0.41$, $p < 0.001$). This dependence was not observed for the corresponding ball velocity coding during the visible epoch (Fig. S7C). This result is noteworthy as the networks that were optimized for dynamic inference were only constrained to explicitly represent the ball position – not velocity.

Taken together, these results point to a critical role of the explicit representation of ball velocity in RNN hidden states, and suggest that establishing autonomous activity dynamics that approximate both ball position and velocity may be a key feature for neural networks to solve M-Pong using dynamic inference-based strategies.

## Discussion

A major hypothesis in cognitive neuroscience is that humans rely on mental simulations to make inferences about latent states of the world[1,3,4,26]. This hypothesis has been implicated in a wide range of behaviors that encompass online versus offline, and open-loop versus closed-loop simulations[5–7,27,28]. In this study, we focus on the potential role of online open-loop simulation in a simple ball interception task. The strongest evidence in support of this type of simulation comes from the observation that high-level computer programs running simulations can emulate human behavior during physical inferences[1,8]. There are, however, critical open questions as to whether and when primates rely on such simulations. First, whether such high-level programs are suitable abstractions for how neural systems compute is debatable[29]. Second, while there has been significant progress in creating model-based neural network agents[30–32], exerting flexible control over such neural models has proven challenging[33], with notable exceptions[34]. Third, somewhat paradoxically, model-free neural network agents that do not rely on mental simulations can outperform their model-based counterparts in rich environments such as Atari games[35]. Together, these considerations highlight the need of revisiting the mental simulation hypothesis using neural network models that afford flexible inferences.

We designed a task that requires humans, monkeys, or artificial models to intercept a ball as it moves behind an occluder. A plausible, but not necessary, strategy to solve this task is to track the position of the ball dynamically as it moves behind the occluder; i.e., perform online open-loop simulations. We compared primate behavior to RNNs that were either only trained to perform the task, or trained additionally to dynamically track the latent state of the ball. Networks that were not optimized for dynamic inference were able to solve the task by finding an "automatized" nonlinear function that mapped sensory inputs to a suitable final paddle position. This finding corroborates recent advances in AI showing neural networks' capacity to implement arbitrarily complex input-output mappings[17,36,37]. Networks that were optimized for dynamic inference were also able to attain primate-level performance, and by construction, carried an internal representation of ball position. Importantly, however, the patterns of errors primates made while performing the task were highly structured and were only captured by RNNs that were additionally capable of simulating the ball position. This finding was not limited to specific parameterization of RNN models, but robust across hundreds of RNNs with different units, inputs, and architectures (Fig. S5A), and did not reflect spurious correlations, but rather causal dependencies (Fig. S6C).

Why do RNNs endowed with dynamic inference ability exhibit error patterns similar to those of primates? We found that humans and monkeys tended to produce larger errors in conditions where the vertical ball displacement during the occluded epoch was larger (Fig. S3), consistent with a noisy biased simulation with a prior towards the ball's last visible vertical position (Fig. S4C). Concurrently, we found that RNN models that exhibited a similar property also had the highest consistency scores, with respect to both human and monkey behavior (Fig. S5C). These results suggest that the similarity in error patterns between RNNs and primates may stem from an optimization of behavior in the context of a common underlying prior, which RNNs optimized for dynamic inference learned from the statistics of their training data.

Our results build on the general framework of using machine learning approaches[38] to create neural networks that successfully model behavior[39–41]. However, unlike research in the sensory systems in which networks with superior performance are also superior in capturing behavior[11,12], we found that network performance was not a good predictor of resemblance to primate behavior. Instead, the networks that most successfully capture primate behavior were those that were additionally constrained to perform dynamic inference. This approach may be understood as a generalization of simultaneously

optimizing on multiple tasks[42,43], or of optimizing for specific tasks in the face of specific regularization[14,44], with the goal of building interpretable models of behavioral and neural phenomena[45]. To this end, our work highlights a general approach for testing hypotheses about specific inductive biases that govern human cognition by directly comparing models that do or do not implement those biases. We used this approach to test the role of "mental simulations" in making inferences, but the same logic can be applied to other hypothesized building blocks of cognition such as hierarchical information processing[46] and counterfactual reasoning[47].

Our modeling efforts can be improved in several ways. First, the RNNs presented here lack many potentially relevant features of biological neural networks (e.g., spiking activity, cell types, architectural constraints), and can only be compared to neurons in the brain at the level of firing rates unless additional biological constraints are imposed[43,48,49]. Second, the sensory input to our models did not incorporate the sensory feedback about the position of the paddle throughout the trial, which humans and monkeys could potentially use to reduce movement variability[50]. However, this is unlikely to be a concern in our work since RNNs did not exhibit this type of output variability (RNN units were not noisy). Moreover, with further analysis, we verified that RNNs did have access to an internal feedback signal related to the moment-by-moment position of the paddle through recurrent connections (Fig. S7D). In other words, RNNs could indeed 'see' and make use of the paddle position through their internal dynamics. In spite of these implementation differences, we were able to discover specific models that predict human behavior to near human-monkey consistency, suggesting that these models reflect specific task-relevant computations in the primate brain. These specific RNN models serve as pre-registered instantiations of neural hypotheses that can be directly tested with future recording from the primate brain.

One desirable feature of RNN models is that they can be used to generate specific and testable hypotheses for the underlying computations in the brain[24,51]. To better understand the differences between the RNNs, we analyzed the structure of their internal state dynamics. We observed that the RNN state dynamics were generally simpler (i.e., slow and smooth) for RNNs that were optimized to simulate the ball position[25]. This is unsurprising given that the ball position changed smoothly[52]. Based on this observation and several recent reports of task-relevant slow dynamics in the primate brain[13,14,16], we wondered if the slow dynamics was the main factor for emulating primate-like behavior. To test this possibility, we analyzed the behavior of a new batch of RNNs that were optimized to perform the task and exhibit simple dynamics but not constrained to simulate the ball position. These networks were also able to perform the task but failed to capture primate behavior as well as RNNs optimized for dynamic inference. Taken together, these results suggest that the primate brain solves the task by establishing slow dynamics that manifest an internal model of the ball position.

We further analyzed the internal dynamics of RNNs asking how the visual input early in the trial enables networks to simulate ball position later in the occluded portion of the trial. Since updating ball position depends on knowledge about velocity, we quantified the degree to which different RNNs carried information about velocity. The velocity information was stronger and more readily decodable (via linear decoders) in the subset of RNNs that were optimized for simulating ball position (Fig. 5). This result highlights an additional signature of dynamic inference in neural networks and serves as a prediction for future physiology experiments.

Previous research on the neurobiology of physical inference and mental simulation has been largely limited to neuroimaging experiments in humans. Due to the inherent limitations of such non-invasive techniques, work in humans has only been able to delineate the neural basis of physical inferences at a macroscopic scale[53,54]. To gain a detailed understanding of the underlying neural circuits and mechanisms, it is important to establish a suitable animal model for mental simulation. Crucially, a suitable animal model would not require over-training (with tens of thousands of repeats of the same stimulus-response contingencies) to capture this behavior, as this could generate alternative behavioral strategies (e.g., "automatized" stimulus-response policies or "memorization") and corresponding spurious underlying neural strategies. Our work establishes such a suitable animal model. First, our head-to-head comparisons of performance indicated that monkeys' error patterns were virtually identical to humans. Second, monkeys could rapidly learn this task and could readily generalize, i.e., adapt properly to previously unseen data drawn from the same distribution as the one used for training. Finally, the simplicity of the M-Pong makes it possible for future work to examine more advanced out-of-domain generalizations involving new objects and new object dynamics. As such, our work establishes a platform for further validation of the online simulation hypothesis and detailed characterization of its underlying neural mechanisms.

## Methods

Two adult monkeys (Macaca mulatta; female), and twelve human participants (18–65 years, gender not queried) participated in the experiments. The Committee of Animal Care and the Committee on the Use of Humans as Experimental Subjects at Massachusetts Institute of Technology approved the animal and human experiments, respectively. All procedures conformed to the guidelines of the National Institutes of Health.

### Behavioral task (M-Pong)

In *M-Pong*, the player controls the vertical position of a paddle along the right edge of the screen to intercept a ball as it moves rightward. On each trial, the ball starts at a random initial position $(x_0, y_0)$ and a random initial velocity $(dx_0, dy_0)$, and moves at a constant speed throughout the trial. The screen additionally contains a large rectangular occluder right before the interception point such that the ball's trajectory is visible only during the first portion of the trial. Trial conditions were constrained by the following criteria: (1) the ball always moved rightward ($dx > 0$), (2) the duration of the visible epoch was within a fixed range ([15,45] RNN timesteps or [624.9, 1874.7] ms), (3) the duration of the occluded epoch was within a fixed range ([15,45] RNN timesteps or [624.9, 1874.7] ms), (4) the number of times the ball bounced was within a fixed range ([0,1]). These constraints imposed some covariations between the ball parameters (e.g., trials, where the ball started farther from the occluder, tended to also have greater ball speed), as shown in Fig. S1. We sampled up to 212480 unique conditions for RNN training (Fig. S1A), and 200 held-out conditions for testing RNNs, humans, and monkeys (see Fig. S1C). Stimuli and behavioral contingencies were controlled by an open-source software (MWorks; http://mworks-project.org/) running on an Apple Macintosh platform.

### Experimental procedures

**Humans.** We collected behavioral data from 12 human participants each performing 1 h of M-Pong behavior. Participant genders were 6 female, 5 male, 1 undisclosed; other population characteristics (e.g., age) were not requested. Participants provided informed consent before the study.

Participants were seated in front of a computer in a dark room, under soft head restraint using a chin-rest. Stimuli were presented on a fronto-parallel 23-inch display (distance: approximately 67 cm; refresh rate: 60 Hz; resolution: 1920 by 1200) and behavioral responses were registered using a standard Apple keyboard. Eye position was tracked every 1 ms with an infrared camera (Eyelink 1000; SR Research Ltd, Ontario, Canada). Each trial was initiated when the participant acquired and held gaze on a central fixation point (white circle,

diameter: 0.5 degrees in visual angle in size) within a window of 4 degrees of visual angle for 200 ms. Following this fixation acquisition, participants were allowed to make eye movements and freely view the screen (see Fig. S8A, B). Afterwards, the M-Pong condition was rendered onto the screen with the entire frame spanning 20 degrees of visual angle: the ball was rendered at its initial position (x0, y0), and the paddle was rendered in the central vertical position at the right edge of the frame. As shown in Fig. 1C, the paddle was initially rendered as a small, transparent green square (0.5 deg × 0.5 deg), but turned into a full paddle (0.5 deg x 2.5 deg) when the participant first initiated paddle movements (i.e., pressed a key). This feature enforced participants to move the paddle on all trials. For the remainder of the trial, participants could freely view the monitor as the ball moved at its fixed velocity (dx0, dy0), and move the paddle up or down using a standard computer keyboard. The paddle position was updated on every screen refresh (i.e., every 16.6 ms), and moved at a constant speed of 0.17 deg/16 ms = 0.01 deg/ms. Trial ended when the ball reached the right-end of the screen. At the end of the trial, the occluder disappeared to give participants feedback on their performance. If they successfully intercepted the ball, it would bounce off their paddle (see Fig. 1C); if they had failed to intercept it, it would continue its path off the frame. Trials were separated by an inter-trial-interval of 750 ms.

In addition to the occluded condition, we also tested trials of the same M-Pong conditions under partial occlusion (opacity of occluder corresponding to 95%) where participants could use visually-guided strategies to perform the task. Such visible trials were randomly interleaved on 25% of all trials. Data from visible trials were not included in the analyses presented in the main manuscript, but as expected, error on visible trials was significantly lower than on occluded trials (Fig. S2A).

Each participant was tested on 50–100 unique task conditions (i.e., different initial ball position and velocity), in both visible and occluded conditions, all randomly interleaved. Trials from all 12 participants were pooled together to characterize average human behavior over the complete dataset of 200 conditions. Behavioral error patterns were remarkably similar across participants (Fig. S4A). Altogether, we measured 8985 trials (6701 and 2284 under the occluded and visible conditions, respectively).

We additionally collected 2711 trials (2462 and 249 under the occluded and visible conditions, respectively) of behavioral data from two held-out human participants performing the exact same conditions of the same task, with one small difference: the paddle size was not 0.5 × 2.5 deg, but 0.5 × 1.75 deg in size. For the purpose of the current work, this held-out data served as an independent validation of our human behavioral measurements, and were used to estimate a human-to-human consistency ceiling that could be directly compared to monkey-to-human consistency estimates.

**Monkeys.** Before the experiments, animals were implanted with three pins for head restraint using standard procedures (under general anesthesia and using sterile surgical techniques). During the experiments, animals were seated comfortably in a primate chair, and were head-restrained. For training purposes, we first acclimated animals to a 1 degree-of-freedom joystick placed right in front of the primate chair. Next, we started a curriculum for training animals to perform M-Pong. Animals were first trained to use the joystick to control the vertical position of a paddle, and then practiced M-Pong using 200 unique trial conditions. For all experiments, the stimuli were presented on a fronto-parallel 23-inch (58.4-cm) monitor at a refresh rate of 60 Hz. Similar to humans, animals' eye position was tracked every 1 ms with an infrared camera (Eyelink 1000; SR Research Ltd, Ontario, Canada). The joystick voltage output (0–5 V) was converted to one of three states (up: 3–5 V; down: 0–2 V; neutral: 2–3 V), which was used to update the position of the paddle. The paddle position was updated in the exact same manner as was done in human experiments: the paddle position

was updated on every screen refresh (i.e., every 16.6 ms), and moved at a constant speed of 0.17 deg/16 ms = 0.01 deg/ms.

We collected behavioral data over 32 behavioral sessions (monkey P: 12; monkey C: 20). Altogether, we measured 52837 trials (39472 and 13365 under the occluded and visible conditions, respectively) with even sampling from both monkeys (monkey P: 19788, 6716; monkey C: 19684, 6649; under occluded and visible conditions respectively). Behavioral error patterns were remarkably similar across the two monkeys (Fig. S4A), and trials from both monkeys were pooled together to characterize monkey behavior.

### RNN optimization

We constructed different recurrent neural network (RNN) models optimized to perform the same task as humans and monkeys. We trained several hundred RNNs to map a series of visual inputs (pixel frames) to a movement output, where the target movement output corresponded to a prediction of the particular paddle position at a particular time point in order to intercept the ball. Different RNN models varied with respect to architectural parameters (different cell types, number of cells, regularization types, input representation types), and were differently optimized (one of four different target outputs, either with or without dynamic inference). Critically, RNNs were not optimized to reproduce primate behavior, only to solve specific tasks. RNNs were trained using the TensorFlow 1.14 library using standard back-propagation and adaptive hyperparameter optimization techniques[55]; training each RNN took one to two days on a Tesla K20 GPU.

We trained two different RNN architectures: LSTM and GRU (for general methodological references regarding these two architecture types, please see Appendix A of[18]). We trained relatively small RNNs (10 or 20 cells) for relatively long durations (100–500 passes through the entire training set, or epochs). LSTM models had four different gates (input, input modulation, forget, and output gates), while GRU models had two different gates (reset and update gates). Each gate was parameterized by two weight matrices of size $N_{input} \times N_{cells}$ and $N_{cells} \times N_{cells}$ (where $N_{input} = 100$, and $N_{cells} = 10$ or 20). All RNNs parameters were initialized to zero prior to training.

Different classes of RNNs with different visual input representations were tested. Each unique task condition consisted of at most 90 timepoints or frames, and we rendered each frame of each trial as a 100 × 100 grayscale image. Note that we did not render the paddle, whose position is controlled by the output of the model. Given the relatively high dimensionality of this input data in the pixel space (90 × 10,000 for each condition), we compressed it using two different encoding transformations. (1) We reduced the dimensionality of the data in pixel-space using principal components analysis, learning the PCA mapping iteratively using small batches of 32 trials at a time, from a subset of 512 total. (2) We additionally tested a higher-level visual representation based on 3-D Gabor wavelets, mimicking the output of neurons in area MT. This Gabor representation was computed with 3D convolutions of the down-sampled image stream with 16 3D Gabor wavelets with spatial and temporal frequencies matching prior work[56]. We reduced the dimensionality of the data in MT-like space using the same iterative PCA strategy.

Different classes of RNNs were trained with one of four different optimization types, which we code-named as no_sim, vis-sim, all_sim and all_sim2. All RNNs had the same number (7) of output channels, each one being an independent linear read-out from the RNN states. Depending on the optimization type, we varied how the overall loss was computed from these output channels (i.e., which output channels were included in estimating the loss). We defined the overall loss as the average of the time-averaged loss of each output channel, thus weighing each output channel equally. Fig. 2C shows the optimization targets for each type. For all RNN types, one of the output channels (called "movement output") corresponded to the paddle position,

which was optimized to predict only two samples per trial: one consisting of the initial central paddle position, and the second corresponding to the particular paddle position at a particular time point in order to intercept the ball. As shown in Fig. 3C, this was the only loss term for RNNs of the "no_sim" class. For the remaining RNNs, we additionally estimated a loss term from some of the other channels, as the mean squared error between the channel output and a target time-varying signal corresponding to the ball's position (x,y) during specific trial epochs ("vis-sim": visual epoch only; "all_sim": entire trial; "all_sim2": separate channels for visual and occluded epochs). This set of optimization choices explored different computational constraints regarding the specifics of dynamic inference processes. For instance, "vis_sim" corresponds to constraints on the sensory but not the latent computations. On the other hand, "all_sim" and "all_sim2" correspond to shared and independent constraints, respectively, on the sensory and latent computations.

Each RNN architecture was trained with and without regularization on the output read-out weights, using either L1 or L2 norm loss. The weight of the regularization loss was measured at two different strengths (0.01, 0.1). Altogether, there were five possible regularization choices. While regularization generally helped with regards to both performance and human-consistency, we found no meaningful difference between L1 and L2 norm loss regularization.

To summarize, we constructed RNN models that varied with respect to several hyper-parameters: different cell types (rnn_type: LSTM or GRU), number of cells (n_hidden: 10 or 20), input representation types (input: pixel_pca or gabor_pca), and regularization types (reg: L1_0.01, L1_0.1, L2_0.01, or L2_0.1, or none); and were differently optimized (loss_weight_type: no_sim, vis_sim, all_sim, or all_sim2). Fig. S5 shows the effect of each hyperparameter choice on performance metrics (task performance, intermediate state decode performance) and primate consistency (with respect to both human and monkey behavior). From the set of tested models, the RNN model architecture with the highest human-consistency score had the following hyper-parameters: {rnn_type: LSTM, n_hidden: 20, input: pixel_pca, reg: L1_0.1, loss_weight_type: all_sim2}.

To investigate whether our results were robust to the extent of RNN training, we additionally tested the effect of the training dataset used for RNN optimization. To do so, we first selected the RNN model architecture with the highest human-consistency score (see above), and evaluated key RNN metrics (e.g., performance, intermediate state decode performance, consistency to humans and to monkeys) while varying both the number of training epochs and the training data (number of training samples and distribution of training data). To test the latter, we first created a larger dataset of M-Pong trials with more variation in ball speed. We found that these metrics were largely insensitive to such variations in RNN optimization (see Fig. S6A). Together, this suggests that the extent of RNN training was sufficient to converge upon "stable" network solutions, and that our key results and inferences are largely robust of the details of this optimization procedure.

To investigate whether our results were robust to the size of RNNs, we first selected the RNN model architecture with the highest human-consistency score (see above) and varied its architecture, testing networks with 100 units and 200 units, of both LSTM and GRU types. We found that these networks exhibited qualitatively similar results (see Fig. S6B), whereby networks optimized for dynamic inference ability were most primate-like in their behaviors, and human consistency scores were correlated to the metric of dynamic inference ability (intermediate state decode performance, ISDP). Together, this suggests that our key results and inferences are largely robust of the details of network size.

Finally, we investigated whether the gains in human-consistency obtained via optimization for simulation could be explained by inducing slow and smooth dynamics in the RNN hidden states. To do so, we

optimized a set of RNN models on task performance (as in "no_sim" models) but with additional regularization to promote slow and smooth dynamics, as in[14]. Specifically, we added three regularization terms corresponding to the L2-norm of the hidden state activity, as the L2-norm of the derivative of the hidden state activity, and the ratio between these two. These three terms were weighted by three corresponding hyper-parameters, which we swept over a broad range in order to ensure that these regularizations had a significant effect on the learned RNN representations. Fig. S7A shows the distribution of human-consistency as well as various representational metrics (dimensionality, speed, curvature, and norm) for all trained RNN models, grouped by their optimization type.

## RNN testing

With these trained RNNs in hand, we estimated a number of properties to characterize each model. Each trained RNN was tested on the same held-out set of 200 unique conditions that were tested in humans and monkeys. We measured the overall performance as the mean absolute error between the final model output (corresponding to a paddle position) and the ground truth final ball position. Given that not all networks were optimized to produce a read-out of the instantaneous ball position, we estimated their ability to "simulate" by training a linear read-out on the RNN states to predict the instantaneous ball position. This read-out was trained on both the visual and occluded epochs and tested on the occluded epoch, and we used a two-fold cross-validation scheme over conditions and time-points. We then quantified the simulation index as the mean absolute error between the predicted and true time-varying ball position.

## Data analysis

Data analysis was performed in Python, using standard python libraries including numpy, scipy, scikit-learn, pandas, matplotlib, and seaborn. Statistical inferences are made from comparisons of datasets using two-tailed Wilcoxon-Mann-Whitney tests. Statistical inferences are made from Pearson correlations by estimating p-values using exact distributions after verifying for normality. All statistical inferences were made using standard scipy implementations.

## RNN characterization

We characterized each RNN's internal state representation via a number of representational attributes. The state representation consists of a matrix $X$ of size $N_{trials} \times N_{timesteps} \times N_{units}$. We first computed the first and second discrete temporal derivatives $X'$ and $X''$. We estimated normalized trajectory speed via the average absolute first derivative, normalized by the average absolute position. Similarly, we estimated a metric of trajectory curvature via the average absolute second derivative, normalized by the average absolute position. To mathematically define these metrics, we first define, for a matrix $A$ of size $N_{trials} \times N_{timesteps} \times N_{units}$, the norm over units $||A||_k$ and the average over timesteps $\mu(A)_j$ as:

$$||A||_k = \sqrt{\sum_{k=1}^{N_{units}} A_{i,j,k}^2}, \text{ and } \mu(A)_j = \frac{1}{N_{timesteps}} \sum_{j=1}^{N_{timesteps}} A_{i,j,k}. \quad (1)$$

Using this notation, the normalized trajectory speed and curvature metrics correspond to:

$$speed = \mu\left(\frac{\mu(|X'|_k)_j}{\mu(|X|_k)_j}\right)_i \text{ and } curvature = \mu\left(\frac{\mu(|X''|_k)_j}{\mu(|X|_k)_j}\right)_i. \quad (2)$$

We then reshaped the state representation matrix of each RNN, concatenating the trial and timestep dimensions, into $X_{mat}$ of size $N_{trials \times timesteps} \times N_{units}$. We estimated the dimensionality of $X_{mat}$ over all

conditions and over the entire trial via a participation ratio metric, a weighted sum of the eigen-spectrum obtained from PCA.

For each RNN model, we additionally estimated a measure of "feedback control" to characterize the alignment between the read-out weights and the recurrent weights. While RNNs do not receive explicit instantaneous visual feedback, this metric aims to capture the extent to which the output of the network is fed back into its activity. For each trained RNN model and for each RNN gate type, we extracted the matrices of input weights ($N_{input\_D} \times N_{units}$), recurrent weights ($N_{units} \times N_{units}$), and read-out weights ($N_{units} \times 1$).

We then computed a metric of feedback control as the normalized projection of the read-out weights onto each of the recurrent weights (i.e., the dot product of the corresponding unit vectors in weight space), and averaging across weights and gate types. Note that our LSTM models have four different gates (*input, input modulation, forget,* and *output* gates), while GRU models have two different gates (*reset* and *update* gates). We observed that the median amount of feedback control was significantly greater than that expected from random read-out weights, across RNNs (Fig. S7D).

### RNN analysis

We note that the different RNN optimization types do not correspond to mutually exclusive hypotheses, but instead map on to overlapping parts of the hypothesis space, as shown in Fig. 4A. For instance, the set of RNNs that were optimized for both task performance and simulation (e.g., "all_sim" and "all_sim2") form a subset of the set of RNNs optimized on task performance alone ("no_sim"). Similarly, the "all_sim" RNNs form a subset of the set of RNNs optimized on task performance and simple dynamics ("simple_dynamics"). As a result, RNNs constructed from different optimization types were not explicitly required to differ with respect to their attributes. For example, RNNs optimized for simulation ability only during the visible epoch ("vis-sim") could still exhibit strong simulation ability during the occluded epoch, despite not being directly optimized for this characteristic.

Given this potential for overlap between optimization types, we did not focus our analyses on group comparisons of consistency scores between the different optimization types. Instead, we sought to infer whether consistency scores depended on specific RNN attributes, over all RNN models, irrespective of the underlying optimization type. We quantified the strength of this dependence via the proportion of variance ($R^2$) of consistency scores that could be explained by each RNN attribute. Moreover, to account for possible co-variations between RNN attributes and infer the proportion of variance that could be uniquely explained by each RNN attribute, we estimated partial $R^2$ (see Comparison metrics).

### Behavioral metrics

We first quantified a grand average estimate of performance using the mean absolute error (MAE), computed as the absolute difference between the final ball position (the center of the ball at the end of the trial) and the final paddle position (the center of the paddle at the end of the trial), averaged across all trials and all conditions.

To go beyond the summary statistic of global performance, we characterized primate and model behavior on this task using a pattern of errors across conditions. This process consists of mapping the final paddle position from a set of trials X ($N_{trials} \times 1$) and the corresponding ground truth paddle positions Z ($N_{trials} \times 1$) to a pattern of errors $Y_{mu}$ ($N_{cond} \times 1$, where $N_{cond} = 200$). However, we cannot simply measure the error of each trial $i$ as a difference between the final paddle position $X_i$ and the corresponding ground truth position $Z_i$, as this will introduce spurious correlations between otherwise unrelated datasets $X_1$ and $X_2$ (e.g., correlations between $X_1$-Z and $X_2$-Z). To mitigate this, we first computed error patterns as residuals from a linear least squares regression $Y = X - X_{pred}$, where $X_{pred} = m*Z + b$ is the linear least squares fit of X. We then averaged trials of the same condition to obtain the pattern of residuals across conditions $Y_{mu}$. Note that, with this definition of error, Pearson correlations between metrics estimated from two datasets is equivalent to computing a partial correlation between the pattern of endpoint paddle positions across conditions, accounting for the co-varying pattern of ground truth positions.

### Behavioral consistency

To quantify the similarity between humans and a model with respect to a given behavioral metric, we used a measure called the "*human-consistency*" ($\hat{\rho}$) as previously defined[57]. *Human-consistency* is computed as a noise-adjusted correlation of behavioral patterns[58,59]. For each system (model or human), we randomly split all behavioral trials into two equal halves and estimated the behavioral pattern from each half, resulting in two independent estimates of the system's behavioral pattern. We took the Pearson correlation between these two estimates as a measure of the reliability of that behavioral pattern given the amount of data collected, i.e., the split-half internal reliability. To estimate the *human-consistency*, we computed the Pearson correlation over all the independent estimates of the behavioral pattern from the model (**m**) and the human (**h**), and we then divide that raw Pearson correlation by the geometric mean of the split-half internal reliability of the same behavioral pattern measured for each system:

$$\hat{\rho}(\boldsymbol{m},\boldsymbol{h}) = \frac{\varrho(m,h)}{\sqrt{\varrho(m,m)\varrho(h,h)}} \tag{3}$$

Since all correlations in the numerator and denominator were computed using the same amount of trial data (exactly half of the trial data), we did not need to make use of any prediction formulas (e.g., extrapolation to larger number of trials using Spearman-Brown prediction formula). This procedure was repeated 10 times with different random split-halves of trials. Our rationale for using a reliability-adjusted correlation measure for *human-consistency* was to account for variance in the behavioral signatures that arises from "noise," i.e., variability that is not replicable by the experimental condition (image and task) and thus that no model can be expected to predict. In sum, if the model (**m**) is a replica of the archetypal human (**h**), then its expected human-consistency is 1.0, regardless of the finite amount of data that is collected.

### Comparison metrics

In addition to reporting individual model scores with respect to this behavioral benchmark[60], we investigated what specific attributes of models best predicts their human-consistency. We estimated the relative importance of specific attributes using a Pearson correlation. To account for covariations due to other attributes, we also report a partial Pearson correlation, the estimated correlation after regressing out co-varying attributes with a linear least squares regression.

For several analyses, we measured the proportion of variance explained of a high-dimensional signal X by another high-dimensional signal Y using an $R^2$ metric. To estimate $R^2$ in an unbiased manner, we performed the following analysis. We first orthogonalized the matrix X into X' using a PCA preprocessing step. We used 5-fold cross-validation to run a linear regression to predict each dimension of X' from Y. We estimated the proportion of variance as the square of the Pearson correlation $R^2$ from this prediction. Across columns of X', this resulted in a vector of $R^2$ for each dimension; we measured the total variance explained as a weighted sum of this vector, with weights corresponding to the eigenvalues of the covariance matrix of X (i.e., the proportion of variance of X explained by each dimension of X').

### Simulation models

To understand primate behavior, we additionally constructed process models that explicitly "simulated" the moment-by-moment ball position during the occluded epoch in the presence of noise to estimate

the endpoint ball position. Given that the probability distributions over possible ball trajectories cannot readily be described analytically, we used a sampling-based (Monte-Carlo) approach to estimate the distributions.

On each run, the simulator is initialized at the beginning of the occluded epoch with the ball's last visible position $(x_0, y_0)$ and velocity $(dx_0, dy_0)$. On each time-step of occlusion, the instantaneous position $(x_t, y_t)$ is estimated using an instantaneous estimate of the velocity $(dx_t, dy_t)$. The instantaneous $dx_t$ behind the occluder was modeled as a sample from an unbiased noisy Gaussian process, $N(dx_0, \sigma^2_{occ-x})$. The instantaneous $dy_t$ behind the occluder was modeled as a sample from a biased Gaussian process $N(dy_0*w_{bias}, \sigma^2_{occ-y})$. Note that this biased Gaussian process is analogous to the combination of an unbiased Gaussian process with a zero-mean prior. We assumed that the simulation noise to be isotropic and used the same $\sigma^2_{occ}$ for both $\sigma^2_{occ-x}$ and $\sigma^2_{occ-y}$. We additionally introduced additive bounce-specific noise modeled as $N(0, \sigma^2_{bounce})$, inspired by[61]. Thus, the stochastic model is compactly described using three parameters: $\sigma^2_{occ}$, $\sigma^2_{bounce}$, and $w_{bias}$.

For each simulation, we used 100 samples and computed the final vertical position $(y_f)$ as the mean vertical position of samples at the end of the occluder. Thus, for each parameter setting, the model produces an estimate for each of the 200 conditions. We fit the model parameters using the L-BFGS-B algorithm for bound constrained minimization using the implementation in the scipy library. Parameters were bound to $0 < \sigma^2_{occ} < 2.0$, $0 < \sigma^2_{bounce} < 2.0$, and $0 < w_{bias} < 2.0$, and the minimization objective was the mean squared error with respect to the final paddle positions across all 200 tested conditions, separately for humans and monkeys.

### Reporting summary

Further information on research design is available in the Nature Research Reporting Summary linked to this article.

## Data availability

The pre-processed data used to generate the associated figures are available on a public repository (https://github.com/RishiRajalingham/MPongBehavior_public). All main and supplemental figures can be reproduced using the notebooks and raw data stored in this repository. Code for reproducing raw data for RNNs is made available at the same repository. Raw data for human and monkey behavior is not shared due to dataset size and complexity, but will be made available upon request to the corresponding author.

## Code availability

All relevant code, including code to generate the custom M-Pong datasets, to train RNN models, to characterize and compare behavior, to perform all relevant analyses, and to generate the associated figures, are available on a public repository: (https://github.com/RishiRajalingham/MPongBehavior_public).

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

## Acknowledgements

R.R. is supported by the Helen Hay Whitney Foundation. M.J. is supported by NIH (NIMH-MH122025), the Simons Foundation, the McKnight Foundation, and the McGovern Institute.

## Author contributions

R.R. and M.J. conceived the study. R.R. collected the monkey data, A.P. collected the human data. R.R. performed all network analyses. M.J. supervised the project. R.R. and M.J. wrote the manuscript.

## Competing interests

The authors declare no competing interests.
