## [Peer Review File · Nature Communications]

Recurrent neural networks with explicit representation of dynamic latent variables can mimic behavioral patterns in a physical inference taskReviewers' Comments:

Reviewer #1:

Remarks to the Author:

Summary

=====

In this paper, the authors used an intuitive physics task (move paddle to catch ball) to compare the performance of humans, non-human primates, and a zoo of RNN models. The RNNs varied by input, architecture, regularization, and optimization. The main yardstick of comparison was "human consistency". The authors conclude that while many of the RNNs achieve reasonable task performance, human consistency is better explained by networks that are optimized to output the ball's moment-to-moment position. The authors take this to suggest a kind of mental simulation underlies primate behavior in this task.

There's a lot to like about this work. Overall I found the paper well written, technically sound, interesting, and a good addition to a topic of current debate. I believe it will be useful for further studies by both this group and other groups, working on human behavior or modeling or neuroscience.

It's the nature of comments that good ones are brief and more critical ones are longer. Below, please find some comments that may help the authors improve the paper. None of these are major bars for publication, most are in the spirit of friendly advice, or questions for follow-up work. The comments are not given in order of importance.

=====

Comments

=====

1. Lack of 'ideal' model comparison: While I take the authors' point that many other people have designed high-level computer simulations for solving intuitive physics tasks, and that the goal of this work is looking at neural-like implementations, I'm left wondering about how such a high-level simulation / process model would explain the results. How exactly would such an ideal model be implemented, anyway? A 'perfect' simulation model could calculate the end-point of the ball as soon as it is visible and zip the paddle to the right location. A noisy-simulation model might be better served waiting till the ball is occluded, since every visible point provides a data sample that can be used to estimate the location and velocity, but as soon as it is behind the screen no further information is useful. Also, such considerations do not yet touch on things like the effort of moving the paddle, the motor execution, and so on. My point is, there isn't *one* process model but rather a series of possible commitments you could be making for how simulation might solve the task. And the reason I'm harping on this note is that there different process models may account for the errors differently. For example, it seems like people aren't moving the paddle "enough". Why is this? Is it explained by a sub-set of the RNN models, and if so HOW is it explained? Can this be accounted for by a noisy high-level process model or do we NEED the RNN model to explain it? Spelling out the process down into what you think is going on regardless of the RNN then gives you hypotheses for what to look for in the RNN. Moment-by-moment simulation is one ingredient of that process model, but not all of it.

2. Collisions and non-collisions, Smith and Vul: If they're not familiar with it, I think the authors would be interested to read and possibly cite Smith and Vul's work on "sources of uncertainty in intuitive physics" -- <http://www.evullab.org/pdf/tops12009.pdf> -- which involves a very similar task to the one under consideration and comes to the conclusion that people have two distinct sources of uncertainty, one having to do with tracking things as they move freely, and one having to do with accounting for collisions. Given that collisions (bouncing off the wall) are a distinct head-ache for RNN models I think it would potentially be useful to examine how well the different parts of the zoo account for errors specifically driven by collisions, and whether anything like two sources of noise emerge in their

behavior. This is also a good way to test generalization -- train with collisions, test without, or vice versa. I feel prompted to say I'm not one of these authors.

3. Micro: While I agree with the citation in line 62, follow-up work by Zhang, Wu et al. has shown many of these supposed successes do not generalize or match human behavior in other ways (see <https://arxiv.org/pdf/1605.01138.pdf>, and again I'm not one of these authors).

4. Over-training and generalization: The authors mention 'over-training' as a possible concern several times, but it wasn't clear to me how much training that is meant to capture. Would the monkeys training on non-occluded screens for thousands of trials not kind as over-training? And how much generalization is captured by showing trajectories not seen before? Presumably if I trained on 1000 examples of a ball going at an angle of 10 degrees at a speed of blah, and then I see a 'new' trajectory of 11 degrees at a speed of blah+1, it's not that big a difference? A better measure of generalization might be going bottom to top instead of left to right, or introducing a barrier in the middle, or shrinking the goal, or some such 'qualitative' difference that keeps the underlying point the same but changes some of the surface details. Same point goes for the RNN zoo, by the way. If some of these models 'truly' came to discover a dynamic simulation procedure, presumably such changes wouldn't hurt them much, right? Or, if it does hurt them a lot, what needs to be added to not be hurt that much? The discussion also mentions 'generalization' but this isn't a very strong test of it.

5. Micro point: The figures are labeled 1,2,4,5,6. That is, figure 3 is missing (or rather, references to figure 4 should be to figure 3, 5 to 4, 6 to 5). This made the text a bit confusing after for references past figure 2. Easy to change, of course.

6. Comparisons beyond optimization type: Perhaps I missed this in the text, but given that you changed a bunch of dimensions besides optimization type I was looking for a clear statement, figure, and comparison to explain how these dimensions affected things (or not). That is, what was the affect of input, architecture, regularization, number of cells? Figure 4F `_kind_` of seems to show cell type didn't matter but it's hard to say and should just be presented separately.

7. Were any of these hypotheses and analyses pre-registered?

8. At several points you mention 'complex non-linear functions' needed to calculate the ball's final location. At least for non-bouncing this seems like a trivial geometric calculation?

9. At several points statistical statements are made without statistical information. I'm mainly referring to lines 200-208. For example, "We first verified that error patterns were similar between the two monkeys (Figure S3B)", similar in what sense? "And that they could not be explained in terms of the ball's initial position and velocity", not explained how? I don't think referencing supplementary figures is useful here.

10. Micro: is there a reason that an untrained RNN is doing so much worse than shuffled null? Perhaps I don't understand shuffled null but should that be the absolute worst performance?

11. I found the section on "Dynamics Underlying Primate-like Behavior" ultimately confusing. I didn't have clear definitions for "slowness" or "dimensionality" and ultimately it was kind of a wash (or worse than the main point). I suggest either expanding on this more clearly or moving to supplementary.

12. Discussion: I think possible interesting points for discussion or follow up work would be: (A) Since over-training *is* a thing in primates, what accounts for it in your RNN zoo? Is the expectation that you have several RNN networks and over time computation shifts from a dynamic-simulation one to a memorization one, or that a single network (e.g. the ones you've identified as the 'right' ones) over time shifts more to memorization? It seems like you could run artificial experiments on your neural network to generate a hypothesis at least, or explain it as you see fit. (B) How would you expect your

networks to handle and explain cases of short-circuiting simulation that don't have to do with over-training and automating? For example, if you added a long "corridor" in the middle of the screen such that a ball which started moving along the corridor would have to end up at the end of it, you would expect people to move their paddle to it without fail, and without reasoning through a simulation (one can consider either Ernie-Davis style scenarios in which you 'know' the right answer thanks to common-sense rules that short circuit simulation but aren't the same as over-training). Given the large and growing interest in combining simulation with rule-like inference, how would your network account for such a thing, if at all?

To clarify, I don't think you need to solve or account for (A) or (B) or even discuss them, these are simply possible points for discussion your paper brought up for me and are live in the literature right now.

Reviewer #2:

Remarks to the Author:

Rajalingham, Piccato, and Jazayeri introduce a new behavioral paradigm to test whether primates perform dynamic inference to track objects in the world. They show that humans and monkeys are capable of similar performance on this simple task, and then use recurrent neural network models to validate that the primates are likely using dynamic inference to complete the task.

I appreciate a lot what this paper is trying to do. Many groups would design an experimental and behavioral paradigm that suits the practicalities of the model organism studied by the lab. This paper, instead, shows the group taking a more considered approach. First, validating that humans and monkeys both perform similarly at the task. Then, using RNN models to validate that the performance of these primates. These results provide a critical basis to, in the future, confidently use the non-human primate recordings to understand implementation-level neural mechanisms.

I am overall quite positive about this manuscript, though I have some concerns and suggestions for improvement.

Major Comments

1) A distinction is drawn between "dynamic inference" and learning "arbitrary nonlinear functions". The RNNs are presented as a crucial piece of evidence to eliminate the latter hypothesis. Yet, it seems that from behavior alone - especially the generalization on first trials to unseen conditions - shows that the subjects are not likely learning arbitrary functions posthoc and are instead relying at least predominately on dynamic inference. Unless, of course, the authors think that the nonlinear functions support generalization, in which case, are these two hypotheses not essentially equivalent since the nonlinear function is enabling inference? The introduction would benefit from a more careful explanation of why it is critical to distinguish between these two hypotheses, and the discussion could more clearly describe why the RNNs are necessary (and able) to provide evidence against the nonlinear function view.

2) Related to the above, the paradigm as designed seems to be as "easy" as possible from a dynamic inference perspective... relatively small occlusion window, fixed set of conditions, simple linear dynamics, etc. The dynamical inference is only needed during a brief window at the end of the trial, which is a much simpler problem than what would need to be solved by the counter-hypothesis, where an arbitrary nonlinear function approximates the entire trajectory. Did the authors explore task performance with truly random sets of initial conditions? What if the occlusion were large enough that users had to rely only on the initial condition? What if, say, the experimenters manipulated the dynamic environment (e.g. adding a flow field where y velocity increased towards the top and bottom of the screen)? These tweaks would allow one to more clearly demonstrate that subjects have inferred

a complete model of the underlying dynamics.

Moderate Comments

1) There is interesting structure in Figure S2C and S3C. What is the cause of the seeming piece-wise, dual-stage increase in error with increase in time occluded? It also appears to show a weak effect of # bounces and dy_0 on paddle error, with the latter having a rather clear pseudo-parabolic structure despite the failed attempt to fit a linear model to assess significance. Did any of the conditions result in the bounce occurring within the occlusion window? If so, was there any bias towards more error on these conditions? Dynamic inference on a straight line through the occluded period seems that it should be easier than estimating the bounce angle without visual feedback.

2) Line 151: Given that the dynamics can be described with a linear model with relatively few parameters, would you necessarily need a nonlinear function to autonomously approximate the ball trajectory on each condition?

3) Figure 2A: Is this data for a single human or averaged across the cohort? It would be useful to see some examples (if not the complete database) of subject-specific behavioral trajectories in the supplement.

4) Unless I am missing something, I did not see a Figure 3 in my version of the PDF; the figures go from 2 to 4. I also didn't see a reference to Figure 3, so I assume it's merely a numbering error, but this should be rectified.

5) Line 618: I think this section could be clearer about how the conditions were selected for the RNN, monkey, and human experiments. As I interpret it, the RNNs were trained using 212480 unique conditions and then the results reported throughout the paper show held-out testing on 200 new conditions which correspond to the exact conditions given to the monkeys and humans. I think the phrasing could be made a bit more explicit if this is true. If this is correct, do the authors foresee any interpretational challenges given that the RNNs were explicitly taught to generalize in order to be included, but the primates were able to learn the trials?

Minor Comments

1) Figure 2C: It's unclear what the blue and green dots represent... the two monkey subjects? Please specify explicitly in a legend on the panel. Additionally, the caption refers to "red and blue arrows" to show the final performance but the Monkeys on this panel is instead purple

2) Figure 4D: for consistency, it might be best to match the colors of the arrows to the colors used in Figure 2C

3) Line 698: it might be worth adding some general methodological references for LSTMs and GRUs to ensure there is no ambiguity for readers about their definitions and what guided your implementations

Reviewer #3:

Remarks to the Author:

In this study, Rishi Rajalingham, Aida Piccato, and Mehrdad Jazayeri test whether primates use mental simulation for tracking dynamical latent states.

First, let me summarize my understanding of some basic concepts to make sure we are on the same page. When sensory measurements are not available, we must rely solely on our internal model to

infer latent states. When the internal model is dynamic e.g. requires knowledge of the physics, such an inference is usually thought to be carried out via mental simulation. The difficulty with studying mental simulation is that you cannot typically monitor the internal thoughts of participants, especially animals. Instead, we must infer whether they perform mental simulation based on their actions. So it comes down to proving that the specific actions taken by the participants could not have been produced without mental simulation. This is what this paper sets out to prove by comparing primate behavior with several task-optimized RNNs with or without some auxiliary loss.

The paper introduces a paradigm called M-pong in which participants adjust a paddle to intercept a ball that they cannot see for the last 900ms or so. Humans and monkeys exhibited a stereotyped pattern of errors across conditions. The error patterns of RNNs trained to track the ball position were more correlated with primate error patterns than RNNs that were not explicitly trained to do so. Control analyses demonstrate that features like slowness of dynamics are not as good at emulating primate behavior. The authors dig into the RNN representations and show that the RNNs represent ball velocity during the occluded epoch, and argue that this might facilitate mental simulation in this task.

My overall impression is that this is a very important study both because it is brave enough to tackle the hard problem of mental simulation, and in doing so introduces a new paradigm that lays the foundation for neural studies in monkeys. The paper was written clearly and was easy to read. The quantification of results seems to have been well thought out. The overall approach is generally sound and the conclusions seem fair but I have some questions:

(1) What was the rationale behind analyzing only the pattern of end point errors? Sure, one has to start somewhere but the task provides us with a dynamic readout (Figure 2A). I would've thought that the paddle dynamics during the occluded epoch convey richer information to test the mental simulation hypothesis. Was that not explored?

(2) From the methods, it looks like the authors performed eye tracking @ 1ms resolution. Again, I would imagine that this would provide us with rich information to test the mental simulation hypothesis e.g. if participants tracked the invisible ball. Were eye movements looked into?

(3) If participants simulate accurately, then there should be no errors. But simulations cannot be perfect, so there are errors. But why would RNNs trained to simulate (but not other RNNs) be expected to make the same kind of errors as humans? Is it because some trials are harder than others when it comes to tracking latent states during the occluded epoch? It would be good to unpack the logic a little bit because it would help me better interpret this result.

(4) From Figure 2A, it looks like the paddle only moved minimally during the occluded phase. Please tell me if this doesn't make sense, but suppose you were to repeat all your analyses with paddle position taken at the start of the occluded epoch before any mental simulation could have happened, would the conclusions still hold? This might be a tall order unless you already attempted it. But could you possibly check and report the following two things?

- Correlation between errors at the start and errors at the end of the occluded epoch (something like figure 2D)

- Human-to-RNN consistency as a function of hidden-state decodability, but with consistency computed w.r.t the start of the occluded epoch (something like figure 3G).

(5) In the same vein, could you compare the error similarity between fully visible trials and trials with occlusion for monkeys (and humans if available)? I would imagine that only the component of error that is not also present in the visible trials could be attributed to mental simulation. On a general note, I appreciate the care taken by the authors in performing all analyses on the deviation from the regression model rather than raw errors.

(6) The consistency in error pattern across monkeys and humans is really striking. I am very curious

to know about the extreme end point error conditions. Could you possibly show paddle positions and ground truth (like Figure 2A middle panel) for trials corresponding to the condition with most positive, most negative, and near-zero errors?

(7) Please correct me if I am wrong, but the test of mental simulation entails a comparison between networks with and without the ability to infer latent states during occluded epoch i.e. all_sim vs vis_sim. I think that the "no_sim" condition inflates the effect sizes in Figure 4 and 5. I say this because inference during the visual epoch requires no simulation — the participant can see the ball. So "vis_sim" is really a misnomer and should ideally be labelled "no_sim" to avoid confusion about the terminology. And "no_sim" should be labelled something else and excluded from R2 computations.

(8) Why is "all_sim2" worse than "all_sim" in tracking the latent state during occlude epoch? Shouldn't it be better because it has a separate readouts for ball position during occlude vs visible epochs, and consequently no interference?

(9) On a related note, what was the rationale behind using pixel-level inputs? Because ball position can be computed from images using feedforward networks, one might only need recurrence for the occluded epoch. Wouldn't it have been easier to isolate the computation of interest (and also from an RNN training standpoint) if the network was directly fed with ball position during the visual epoch? You don't need to do this, but I am curious.

(10) In the final analysis, isn't it odd that the ball velocity coding increases rapidly precisely when the ball is no longer visible? Wouldn't one expect it to go down because velocity information stops coming into the network? I suppose this means that the coding of velocity becomes less nonlinear and more linear as time passes. Could you please shed some light? The effect is very strong (almost 2x jump in R2), hence the question.

Minor points:

(1) Figure n in the text becomes n+1 in figure captions for n>2.

(2) Why does the analysis in Figure S2C use absolute and not raw paddle errors?

(3) Do results in S2D depend on the choice of parametrization (cartesian vs polar)? Are errors predicted by the distance covered after bouncing before the ball becomes invisible?

Reviewer #1 (Remarks to the Author):

Summary

=====

In this paper, the authors used an intuitive physics task (move paddle to catch ball) to compare the performance of humans, non-human primates, and a zoo of RNN models. The RNNs varied by input, architecture, regularization, and optimization. The main yardstick of comparison was "human consistency". The authors conclude that while many of the RNNs achieve reasonable task performance, human consistency is better explained by networks that are optimized to output the ball's moment-to-moment position. The authors take this to suggest a kind of mental simulation underlies primate behavior in this task.

There's a lot to like about this work. Overall I found the paper well written, technically sound, interesting, and a good addition to a topic of current debate. I believe it will be useful for further studies by both this group and other groups, working on human behavior or modeling or neuroscience.

It's the nature of comments that good ones are brief and more critical ones are longer. Below, please find some comments that may help the authors improve the paper. None of these are major bars for publication, most are in the spirit of friendly advice, or questions for follow-up work. The comments are not given in order of importance.

We thank the reviewer for this positive and accurate summary of our work. We appreciate the reviewer's thoughtful comments and questions for follow-up work. In the revised manuscript, we have tried our best to address all their comments with clarifying text and/or additional analyses and supplemental figures.

=====

Comments

=====

1. Lack of 'ideal' model comparison: While I take the authors' point that many other people have designed high-level computer simulations for solving intuitive physics tasks, and that the goal of this work is looking at neural-like implementations, I'm left wondering about how such a high-level simulation / process model would explain the results. How exactly would such an ideal model be implemented, anyway? A 'perfect' simulation model could calculate the end-point of the ball as soon as it is visible and zip the paddle to the right location. A noisy-simulation model might be better served waiting till the ball is occluded, since every visible point provides a data sample that can be used to estimate the location and velocity, but as soon as it is behind the screen no further information is useful. Also, such considerations do not yet touch on things like the effort of moving the paddle, the motor execution, and so on. My point is, there isn't *one* process model but rather a series of possible commitments you could be making for how simulation might solve the task. And the reason I'm harping on this note is that there different process models may account for the errors differently. For example, it seems like people aren't moving the paddle "enough". Why is this? Is it explained by a subset of the RNN models, and if so HOW is it explained? Can this be accounted for by a noisy high-level process model or do we NEED the RNN model to explain it? Spelling out the process down into what you think is going on regardless of the RNN then gives you hypotheses for what to look for in the RNN. Moment-by-moment simulation is one ingredient of that process model, but not all of it.

The reviewer has asked that we compare human/monkey behavior to RNNs as well as various process models that are optimized for different cost functions. The reviewer highlights that such process models could expose

the computational components (such as mental simulation) that underlie behavior and thus can serve as “hypotheses for what to look for in the RNN.”

We made several revisions to address this comment. First, we carefully analyzed the statistics of the primate behavior with respect to various hypotheses such as the effect of motor cost, accuracy of motor planning, and simulation noise during occlusion. Results indicated that endpoint errors were larger when the ball’s vertical displacement behind the occluder was larger (Fig. S3). This finding is consistent with the accumulation of simulation noise during the occluded epoch. We also evaluated endpoint errors with respect to other hypotheses. For example, we tested whether errors could be related to a simple motor cost (i.e., the distance the paddle had to move). Results were not consistent with this hypothesis as there was no significant correlation between endpoint error and the magnitude of the paddle movement (Fig. S3). We also tested the hypothesis of whether a longer visual epoch could improve motor preparation. Again, the results rejected this hypothesis as endpoint errors did not depend on the duration of the visible epoch (Fig. S3). These analyses indicate that a process akin to mental simulation was the major source of error in behavior.

Next, we developed various process models to examine the role of various noise sources on the observed behavioral statistics (Fig. S4). The models we considered were characterized by three parameters, simulation noise, bounce-induced noise, and a regression weight towards the last visible ball position. We then fit this class of models to human and monkey behavior. Results revealed that both noisy simulation and a bias towards the last visible ball position were important for capturing behavioral statistics. The smallest contribution was from the bounce noise that minimally improved model fits. We added a new supplemental panel (Fig S4C) comparing this model to four meaningful null models, and presenting simulated model outputs against human behavior.

Please note that the bias is a natural consequence of adopting a performance-optimizing Bayesian integration strategy in the presence of simulation noise (i.e., the classic bias-variance trade-off in Bayesian models). The fact that models with this additional bias provided a better fit to behavior indicates that humans and monkeys seek to optimize their responses in the presence of simulation noise, as numerous previous experiments have found. Note however that although our RNN models with simulation ability were able to reproduce these behavioral biases (“endpoint error”), we cannot straightforwardly draw parallels between the various sources of variability in the noisy process models and our RNNs models because the two models are very different in terms of how they implement computations.

We wish to thank the reviewer for encouraging us to make these revisions.

2. Collisions and non-collisions, Smith and Vul: If they’re not familiar with it, I think the authors would be interested to read and possibly cite Smith and Vul’s work on “sources of uncertainty in intuitive physics” -- <http://www.evullab.org/pdf/tops12009.pdf> -- which involves a very similar task to the one under consideration and comes to the conclusion that people have two distinct sources of uncertainty, one having to do with tracking things as they move freely, and one having to do with accounting for collisions. Given that collisions (bouncing off the wall) are a distinct head-ache for RNN models I think it would potentially be useful to examine how well the different parts of the zoo account for errors specifically driven by collisions, and whether anything like two sources of noise emerge in their behavior. This is also a good way to test generalization -- train with collisions, test without, or vice versa. I feel prompted to say I’m not one of these authors.

We thank the reviewer for referring us to this related work. To summarize the reviewer’s point, it is possible that the errors made by humans/monkeys/RNNs can be characterized by separate reflection-dependent and reflection-independent components. In the previous manuscript draft, we showed a coarse comparison of

errors for conditions with and without reflections (Figures S2C and S3C) that suggest that this may be true for primates; we have expanded this in the current revision to separately examine occluded vs visible bounces (Fig S2C). We additionally present this analysis for all tested RNN models, separated by optimization type (Figure S5B). This figure shows that all RNN types exhibit greater errors on average for trials with bounces, suggesting that this coarse behavioral signature does not, by itself, discriminate between alternative RNN optimization types.

Additionally, prompted by the reviewer's suggestion above, we have incorporated this algorithmic hypothesis into a new set of analyses using process models (see response for Point #1 above). Specifically, we constructed noisy simulation models with a bounce-specific source of variance, and fit model parameters to match primate behavior. This analysis revealed a small effect of reflection-specific noise for humans and monkeys (Fig S4C).

Finally, while we appreciate the point about training with collisions and testing without (or vice versa), this effort is bound to fail as has been shown repeatedly in the machine learning literature. Generalization is something that the entire field is grappling with. The problem, at its core, is that these models learn fixed statistical regularities and cannot generalize to out-of-distribution conditions. Bounces present a perfect example of an out-of-distribution scenario that our models as well as any other model without the appropriate built-in inductive biases would fail to generalize to.

3. Micro: While I agree with the citation in line 62, follow-up work by Zhang, Wu et al. has shown many of these supposed successes do not generalize or match human behavior in other ways (see <https://arxiv.org/pdf/1605.01138.pdf>, and again I'm not one of these authors).

We thank the reviewer for this reference; we have added it to the manuscript text.

4. Over-training and generalization: The authors mention 'over-training' as a possible concern several times, but it wasn't clear to me how much training that is meant to capture. Would the monkeys training on non-occluded screens for thousands of trials not kind as over-training? And how much generalization is captured by showing trajectories not seen before? Presumably if I trained on 1000 examples of a ball going at an angle of 10 degrees at a speed of blah, and then I see a 'new' trajectory of 11 degrees at a speed of blah+1, it's not that big a difference? A better measure of generalization might be going bottom to top instead of left to right, or introducing a barrier in the middle, or shrinking the goal, or some such 'qualitative' difference that keeps the underlying point the same but changes some of the surface details. Same point goes for the RNN zoo, by the way. If some of these models 'truly' came to discover a dynamic simulation procedure, presumably such changes wouldn't hurt them much, right? Or, if it does hurt them a lot, what needs to be added to not be hurt that much? The discussion also mentions 'generalization' but this isn't a very strong test of it.

The reviewer tackles an important point here, namely the ill-defined word "generalization." In machine-learning and neuroscience, generalization commonly refers to a model's "ability to adapt properly to new, previously unseen data, drawn from the same distribution as the one used to create the model." However, as the reviewer correctly points out, the details of whether the unseen data are "far" from previously seen data or not adds a level of nuance to the word generalization.

To illustrate this, the reviewer provides an extreme example where animals are over-trained on thousands of conditions and tested on novel conditions that are minimally different from these training conditions. To clarify, this is **not** the regime used in our animal behavior, and we have added a supplemental figure that describe the regime of training conditions (50 unique conditions, sampled randomly from the full space of ball

position/velocity parameters) and held-out test conditions (150 unique conditions, sampled randomly from the full space of ball position/velocity parameters). Importantly, it is not clear how to characterize the task conditions: e.g. using the initial position/velocity of the ball, or the time-course of ball position, or even the time-course of pixel inputs. In this revision, we chose a simple characterization (four parameters for initial ball position and velocity), and provided the reader/reviewer with histograms of all pairwise joint distributions, highlighting train and test samples.

The notion of generalization can also be extended to out-of-domain generalization, which measures the models' ability to adapt to unseen data drawn from a different distribution, such as the kinds of tests that the reviewer suggests (e.g. introducing new objects, new object dynamics). We completely agree that these kinds of generalizations are interesting for future work, especially given that out-of-domain generalizations appear to be a capability of humans, but remain **the most difficult problem for current machine learning models**. We hope the reviewer agrees that solving out-of-domain generalization is not a requirement for supporting the inferences made in our current work. We have instead edited the text to clarify the notion of generalization as used here, and discuss the nuances brought up by this comment.

Minor point of clarification: Regarding the question of whether “training on non-occluded screens for thousands of trials [is] not kind as over-training”, Figure 2C demonstrates that our estimate of “generalization performance did not reflect prior exposure to the corresponding visible conditions, ... as evidenced by the similar performance on the subset of test conditions with no such prior exposure.” In other words, the generalization was not only to occluded vs non-occluded, but simultaneously to novel task conditions. We have edited the text to make this explicit.

5. Micro point: The figures are labeled 1,2,4,5,6. That is, figure 3 is missing (or rather, references to figure 4 should be to figure 3, 5 to 4, 6 to 5). This made the text a bit confusing after for references past figure 2. Easy to change, of course.

We apologize for this typo, and thank the reviewer for finding it! We have fixed it in the revision.

6. Comparisons beyond optimization type: Perhaps I missed this in the text, but given that you changed a bunch of dimensions besides optimization type I was looking for a clear statement, figure, and comparison to explain how these dimensions affected things (or not). That is, what was the affect of input, architecture, regularization, number of cells? Figure 4F _kind_ of seems to show cell type didn't matter but it's hard to say and should just be presented separately.

We thank the reviewer for this point. In the last submission, we had included Figure S4C to present the effect of architectural hyper-parameters. This four-panel figure shows the difference in several outcome metrics (performance, dynamic inference ability, consistency to humans, and consistency to monkeys) as a function of the network size, cell type, input representation, and output regularization. This figure may combine too many dimensions to effectively parse it at a glance, so we have reformatted it into a 16 panel figure that shows the effect of each dimension individually.

7. Were any of these hypotheses and analyses pre-registered?

We did not pre-register our hypotheses or analyses. That said, we see our RNN models as concrete neural hypotheses. Therefore, results from this work serve as pre-registration of specific hypotheses for our ongoing work involving recording from the primate brain. With these hypotheses published, there is now no room for us

to engage in post-hoc hypothesis generation and testing for the physiology data, at least not without acknowledging the failures.

8. At several points you mention 'complex non-linear functions' needed to calculate the ball's final location. At least for non-bouncing this seems like a trivial geometric calculation?

We thank the reviewer for pointing out this need for clarification. As the reviewer surmises, extrapolation of the ball position along a straight path is trivial when the problem is stated in terms of appropriate latent variables such as initial position (x_0, y_0) and velocity (dx, dy). However, any model including our RNNs has to first infer these latent variables from the pixel input, which involves nonlinear computations. Moment-by-moment updating of the simulation also involves highly nonlinear computations since the moment-by-moment sensory feedback is in the pixel domain and not in the form of the desired latent variables. However, we agree with the reviewer that the task demands are relatively simple for human observers, and we have removed the term "complex" in two relevant occurrences in the text (line 145, and line 150).

9. At several points statistical statements are made without statistical information. I'm mainly referring to lines 200-208. For example, "We first verified that error patterns were similar between the two monkeys (Figure S3B", similar in what sense? "And that they could not be explained in terms of the ball's initial position and velocity", not explained how? I don't think referencing supplementary figures is useful here.

We thank the reviewer for pointing out these omissions. We have now included this statistical information in the main text.

10. Micro: is there a reason that an untrained RNN is doing so much worse than a shuffled null? Perhaps I don't understand shuffled null but should that be the absolute worst performance?

To clarify, the shuffled null does not correspond to the absolute worst, as it is obtained by shuffling the correct outputs (y_t) for each condition, and thus preserves the distribution of y_t . Untrained RNNs do not have access to this prior distribution, and can only learn it from data. We have edited the text to clarify this.

11. I found the section on "Dynamics Underlying Primate-like Behavior" ultimately confusing. I didn't have clear definitions for "slowness" or "dimensionality" and ultimately it was kind of a wash (or worse than the main point). I suggest either expanding on this more clearly or moving to supplementary.

We think that keeping this section in the main manuscript is critical in light of prior work. Recent studies have suggested that RNNs with "simple" dynamics are more similar to neural data (e.g., in the context of movement and motor planning). Indeed, in presenting this work to various communities, we were often asked whether the results can be explained in terms of slowness and dimensionality of the internal dynamics. We found that these factors alone are unable to account for why RNNs with simulation capability are more primate-like in their behavior. However, we agree that it is important to clarify what we mean by simple dynamics. We have attempted to clarify this section and included those definitions in the Results section.

12. Discussion: I think possible interesting points for discussion or follow up work would be: (A) Since over-training *is* a thing in primates, what accounts for it in your RNN zoo? Is the expectation that you have several RNN networks and over time computation shifts from a dynamic-simulation one to a memorization one, or that a single network (e.g. the ones you've identified as the 'right' ones) over time shifts more to memorization? It seems like you could run artificial experiments on your neural network to generate a hypothesis at least, or explain it as you see fit. (B) How would you expect your networks to handle and explain

cases of short-circuiting simulation that don't have to do with over-training and automating? For example, if you added a long "corridor" in the middle of the screen such that a ball which started moving along the corridor would have to end up at the end of it, you would expect people to move their paddle to it without fail, and without reasoning through a simulation (one can consider either Ernie-Davis style scenarios in which you 'know' the right answer thanks to common-sense rules that short circuit simulation but aren't the same as over-training). Given the large and growing interest in combining simulation with rule-like inference, how would your network account for such a thing, if at all?

To clarify, I don't think you need to solve or account for (A) or (B) or even discuss them, these are simply possible points for discussion your paper brought up for me and are live in the literature right now.

We thank the reviewer for these comments regarding potential future work. To summarize, the reviewer is proposing the idea that (over)training drives a shift from slow, deliberate cognitive strategies to rapid, automated memorization-like strategies. This is definitely a compelling proposal for behavior in general, and one that is of great interest to us for ongoing and future work. We think about this especially in the context of discrepancies in behavioral strategies for different tasks that are tested in the field. However, we believe this discussion point is largely beyond the scope of this current paper and would be overly speculative (i.e., we have no results to illuminate the ongoing rich discussions on these topics).

Reviewer #2 (Remarks to the Author):

Rajalingham, Piccato, and Jazayeri introduce a new behavioral paradigm to test whether primates perform dynamic inference to track objects in the world. They show that humans and monkeys are capable of similar performance on this simple task, and then use recurrent neural network models to validate that the primates are likely using dynamic inference to complete the task.

I appreciate a lot what this paper is trying to do. Many groups would design an experimental and behavioral paradigm that suits the practicalities of the model organism studied by the lab. This paper, instead, shows the group taking a more considered approach. First, validating that humans and monkeys both perform similarly at the task. Then, using RNN models to validate that the performance of these primates. These results provide a critical basis to, in the future, confidently use the non-human primate recordings to understand implementation-level neural mechanisms.

I am overall quite positive about this manuscript, though I have some concerns and suggestions for improvement.

We thank the reviewer for this positive and accurate summary of our work.

Major Comments

1) A distinction is drawn between “dynamic inference” and learning “arbitrary nonlinear functions”. The RNNs are presented as a crucial piece of evidence to eliminate the latter hypothesis. Yet, it seems that from behavior alone - especially the generalization on first trials to unseen conditions - shows that the subjects are not likely learning arbitrary functions post hoc and are instead relying at least predominately on dynamic inference. Unless, of course, the authors think that the nonlinear functions support generalization, in which case, are these two hypotheses not essentially equivalent since the nonlinear function is enabling inference? The introduction would benefit from a more careful explanation of why it is critical to distinguish between these two hypotheses, and the discussion could more clearly describe why the RNNs are necessary (and able) to provide evidence against the nonlinear function view.

We thank the reviewer for bringing up this point of confusion. The reviewer mentions two different ideas here, 1) dynamic inference vs. unconstrained nonlinear function, and 2) generalization vs over-fitting. It is important to note that these are orthogonal ideas: models can generalize well using either a dynamic inference strategy (i.e. by performing a moment-by-moment tracking of the latent ball) or using an alternative strategy (e.g. by estimating the final ball position using a static nonlinear geometric computation from only the initial ball state, which does not require moment-by-moment tracking). Critically, these different strategies, while both capable of generalizing to novel conditions, will exhibit different patterns of errors. The goal of this current work is to use the measured patterns of errors to infer which of these two strategies is most consistent with primate behavior. As described in the Introduction, the Mental-Pong task “is particularly suitable for testing the dynamic inference hypothesis because, in principle, it can be solved both by a dynamic inference engine and an automatized nonlinear function. In other words, dynamic inference is not an obligatory consequence of task design but rather a solution that primates might plausibly adopt.” We have edited the text to further clarify this point in the Introduction.

2) Related to the above, the paradigm as designed seems to be as “easy” as possible from a dynamic inference perspective... relatively small occlusion window, fixed set of conditions, simple linear dynamics, etc.

The dynamical inference is only needed during a brief window at the end of the trial, which is a much simpler problem than what would need to be solved by the counter-hypothesis, where an arbitrary nonlinear function approximates the entire trajectory. Did the authors explore task performance with truly random sets of initial conditions? What if the occlusion were large enough that users had to rely only on the initial condition? What if, say, the experimenters manipulated the dynamic environment (e.g. adding a flow field where y velocity increased towards the top and bottom of the screen)? These tweaks would allow one to more clearly demonstrate that subjects have inferred a complete model of the underlying dynamics.

The reviewer highlights a number of fascinating follow-ups to our work. We are excited to extend our work to test all these follow-ups and more (e.g., addition of teleportation portals, multiple balls, etc.). However, our choice of starting from “easy” was intentional. Since we have no a-priori hypothesis about how the primate brain solves such tasks, we think it is prudent to start from “easy” and test the internal dynamics of the models we have so far developed against brain activity. In essence, we don’t want to go too far down a certain branch of the model tree based on behavioral generalizations alone and without any validation from the underlying neural data. We are currently recording from monkeys’ frontal cortex to directly test the simulation hypothesis.

We expect to find discrepancies between the internal dynamics of our current models and the recordings from the primate brain. We hope to exploit those discrepancies to make more educated choices about the subsets of “tweaks” that we should consider for arbitrating between the next generation of models. Regardless, we have revised the manuscript to highlight the importance of exploring various dimensions of generalization in future studies in the Discussion.

Minor points of clarification: The initial conditions used here were “truly random” with only the constraints stated in Methods lines 610-615. We also used a fixed set of 200 conditions, which we have previously observed is a sufficiently large number of conditions that humans do not realize it is fixed (when asked directly after the experiment).

Moderate Comments

1) There is interesting structure in Figure S2C and S3C. What is the cause of the seeming piece-wise, dual-stage increase in error with increase in time occluded? It also appears to show a weak effect of # bounces and dy_0 on paddle error, with the latter having a rather clear pseudo-parabolic structure despite the failed attempt to fit a linear model to assess significance. Did any of the conditions result in the bounce occurring within the occlusion window? If so, was there any bias towards more error on these conditions? Dynamic inference on a straight line through the occluded period seems that it should be easier than estimating the bounce angle without visual feedback.

The reviewer alludes to interesting structure in Figures S2C/S3C. In this revision, we show that errors depended most strongly on the vertical ball displacement during the occluded epoch (see Figure S3), which is approximately the product of the time occluded and the vertical ball velocity (before accounting for bounces). In other words, the apparent “dual-stage” relationship between time occluded and error is due in large part to covariations in vertical ball velocity. This dependence is consistent with a noisy simulation model wherein errors in dynamic inference accumulated on a moment-by-moment basis during the occluded epoch.

Additionally, the reviewer reasons that conditions in which the ball bounced behind the occluder should exhibit greater error, relative to all other conditions. We examined this in a new supplemental figure panel (Fig S2C). Given that there are a relatively small number of these conditions, the *specific* effect of occluded bounces is difficult to infer from this, but we observe that conditions with occluded bounces have a larger range of errors.

To directly examine the effect of occluded bounces, we constructed and fit process models that incorporate a bounce-specific source of noise. These results, which we have added to supplemental figure Fig S4C, demonstrate that human behavior is most consistent with process models that include a small amount of additional simulation noise in the presence of occluded bounces.

2) Line 151: Given that the dynamics can be described with a linear model with relatively few parameters, would you necessarily need a nonlinear function to autonomously approximate the ball trajectory on each condition?

We thank the reviewer for pointing out this need for clarification. In addressing this comment, it is important to distinguish between two different ways an agent can solve the task. Recall that the task is not to track the ball on a moment-by-moment basis (simulation); instead, the subjects only need to compute the endpoint of the ball after exiting the occluder.

Subjects can solve this task in two ways. The first solution is to skip the dynamic tracking altogether and directly compute the endpoint based on initial positions (x_0, y_0) and the ball's velocity (dx, dy). Even when the ball takes a straight path (no bounces), the relationship between the vertical endpoint (y_{end}) and these variables is indeed nonlinear (note the dx in the denominator):

$$y_{end} = y_0 + (dy/dx) (x_{end} - x_0)$$

The situation gets worse (more complex nonlinearities) if we consider ball bounces. Furthermore, nonlinearities are unavoidable if we take into account the fact that trial-specific variables (x_0, y_0, dx, dy) are latent and have to be computed using nonlinear functions of the pixel input. Therefore, direct computation of y_{end} from pixel input depends on nonlinearities.

The second solution is to compute y_{end} by performing simulations. This solution also needs certain nonlinear computations to infer trial-specific latent variables (x_0, y_0, dx, dy) from pixel input. However, after computing these latent variables, the problem can be solved relatively easily. For trials without a bounce, computing y_{end} can be done incrementally (simulation) with a simple linear updating rule (incrementing the x and y position by the corresponding speed). For trials with a bounce, the simulator has to invert the speed along the y dimension at the time of the bounce. This is, of course, a nonlinear operation because it is time-dependent, but it is a point nonlinearity that can be invoked only at the time of the bounce.

In sum, both solutions involve nonlinear computations. Direct computation of the endpoint without simulation involves complex nonlinearities that have to simultaneously account for (1) pixel inputs, (2) time-to-contact, and (3) bounces. The simulation model, in contrast, can limit the nonlinear computations to a front-end mapping of pixel inputs to the relevant latent variables and an additional updating of the speed if/when a bounce happens.

3) Figure 2A: Is this data for a single human or averaged across the cohort? It would be useful to see some examples (if not the complete database) of subject-specific behavioral trajectories in the supplement.

The data corresponds to average across subjects (noted in Methods). Of course, we will make the full dataset available so that others can examine the data for individual subjects. We can include a plot of every trial for every subject in the manuscript but, for multiple reasons, we think that such a plot would not be informative. First, a plot of all trials for every subject would be highly cluttered and difficult to make sense of. Second, it would be difficult to do any statistics on individual subjects because of the small number of repeats per condition per subject (~2.8 trials on average). Third, head-to-head comparison between subjects is not

possible because subjects were tested on non-overlapping subsets of the full 200 conditions (as noted in Methods).

Regardless, we think it is important to provide evidence of overall consistency among subjects. Since the number of unique trial repeats for individual subjects were small, we tested inter-subject consistency by bootstrapping across subjects. Specifically, we compared similarity between two disjoint datasets, each obtained by pooling 6 human subjects. We note that, in the presence of inter-subject variability, this similarity score is dependent on the number of subjects pooled together in each data split. To take this point into account, we first generated predictions for the expected value of inter-subject similarity as a function of the number of subjects for different ground truth values of inter-subject similarity. We then plotted the observed inter-subject similarity for both humans and monkeys relative to this expectation (Fig. S4A). Results suggest that both humans and monkeys have relatively high inter-subject similarity, with inferred correlations between individual subjects of $r = 0.75$ to 0.8 .

4) Unless I am missing something, I did not see a Figure 3 in my version of the PDF; the figures go from 2 to 4. I also didn't see a reference to Figure 3, so I assume it's merely a numbering error, but this should be rectified.

We apologize for this typo. We have fixed it in the revision.

5) Line 618: I think this section could be clearer about how the conditions were selected for the RNN, monkey, and human experiments. As I interpret it, the RNNs were trained using 212480 unique conditions and then the results reported throughout the paper show held-out testing on 200 new conditions which correspond to the exact conditions given to the monkeys and humans. I think the phrasing could be made a bit more explicit if this is true. If this is correct, do the authors foresee any interpretational challenges given that the RNNs were explicitly taught to generalize in order to be included, but the primates were able to learn the trials?

The reviewer is correct that RNNs were trained using up to 212480 unique conditions, and tested on the same held-out set of 200 unique conditions that were tested in humans and monkeys." We have now edited the main text to present this information in the Results section.

The reviewer points out that RNNs were tested on 200 novel conditions (without any supervision on those conditions), while humans and monkeys were repeatedly tested on the same 200 conditions with reinforcement. To our understanding, the reviewer is concerned that this reinforcement could lead to learning or "memorization" of the correct answer in primates. To examine this, we showed that monkeys exhibit the same performance even on conditions they have never previously seen (Fig 2C), which is inconsistent with a memorization strategy. Furthermore, while RNNs and primates certainly have different training histories (e.g. primates have prior experience with moving objects and tasks involving time-to-contact, whereas RNNs have supervised training on a large training set of M-Pong conditions), we note that the training regime was identical across RNNs. Given that our inferences are entirely drawn from comparisons across different RNN models, we do not foresee any issues with our interpretations.

Minor Comments

1) Figure 2C: It's unclear what the blue and green dots represent... the two monkey subjects? Please specify explicitly in a legend on the panel. Additionally, the caption refers to "red and blue arrows" to show the final performance but the Monkeys on this panel is instead purple

We thank the reviewer for catching these typographical mistakes! We have fixed them in the revision.

2) Figure 4D: for consistency, it might be best to match the colors of the arrows to the colors used in Figure 2C

We thank the reviewer for this suggestion, which we have included in the revision.

3) Line 698: it might be worth adding some general methodological references for LSTMs and GRUs to ensure there is no ambiguity for readers about their definitions and what guided your implementations

We thank the reviewer for this suggestion, which we have included in the revision.

Reviewer #3 (Remarks to the Author):

In this study, Rishi Rajalingham, Aida Piccato, and Mehrdad Jazayeri test whether primates use mental simulation for tracking dynamical latent states.

First, let me summarize my understanding of some basic concepts to make sure we are on the same page. When sensory measurements are not available, we must rely solely on our internal model to infer latent states. When the internal model is dynamic e.g. requires knowledge of the physics, such an inference is usually thought to be carried out via mental simulation. The difficulty with studying mental simulation is that you cannot typically monitor the internal thoughts of participants, especially animals. Instead, we must infer whether they perform mental simulation based on their actions. So it comes down to proving that the specific actions taken by the participants could not have been produced without mental simulation. This is what this paper sets out to prove by comparing primate behavior with several task-optimized RNNs with or without some auxiliary loss.

The paper introduces a paradigm called M-pong in which participants adjust a paddle to intercept a ball that they cannot see for the last 900ms or so. Humans and monkeys exhibited a stereotyped pattern of errors across conditions. The error patterns of RNNs trained to track the ball position were more correlated with primate error patterns than RNNs that were not explicitly trained to do so. Control analyses demonstrate that features like slowness of dynamics are not as good at emulating primate behavior. The authors dig into the RNN representations and show that the RNNs represent ball velocity during the occluded epoch, and argue that this might facilitate mental simulation in this task.

My overall impression is that this is a very important study both because it is brave enough to tackle the hard problem of mental simulation, and in doing so introduces a new paradigm that lays the foundation for neural studies in monkeys. The paper was written clearly and was easy to read. The quantification of results seems to have been well thought out. The overall approach is generally sound and the conclusions seem fair but I have some questions:

We thank the reviewer for this positive and accurate summary of our work!

(1) What was the rationale behind analyzing only the pattern of end point errors? Sure, one has to start somewhere but the task provides us with a dynamic readout (Figure 2A). I would've thought that the paddle dynamics during the occluded epoch convey richer information to test the mental simulation hypothesis. Was that not explored?

Our main rationale for focusing on the endpoint error was that the only task requirement in M-Pong is to minimize this error. In other words, the only constraint that all subjects and models had to adhere to was to minimize the endpoint error. Therefore, it seemed reasonable to use this as our primary metric for comparison.

However, we agree that 'the pattern of errors between the paddle trajectory and the ball trajectory during the entire occluded epoch' provide an useful behavioral metric to further compare the models. We performed the analyses suggested by the reviewer. The results were consistent with our original inference that the RNN models with simulation were more consistent with the primate behavior. These results are documented in a new supplementary figure (Fig. S9B).

(2) From the methods, it looks like the authors performed eye tracking @ 1ms resolution. Again, I would imagine that this would provide us with rich information to test the mental simulation hypothesis e.g. if participants tracked the invisible ball. Were eye movements looked into?

We are sympathetic to this suggestion but we note that unrestrained eye movements can be idiosyncratic and highly heterogeneous and thus very difficult to analyze. We have included a supplemental figure that shows example eye traces for different conditions (Figure S8A), and coarsely characterized the dynamics of gaze position during the visible and occluded epochs (Figure S8B) to illustrate the complexity of these data.

The reviewer highlights a specific hypothesis to test: “[do] participants tracked the invisible ball[?]” We cannot test this hypothesis because humans and monkeys are unable to generate smooth pursuits to track invisible moving objects (except in rare and highly constrained laboratory experiments). Therefore, we cannot use the eye position to directly test for the simulation hypothesis. In lieu of such analysis, we analyzed the subjects’ gaze positions at specific time points during the occluded epoch (final, last-occluded, and mid-occlusion) relative to the corresponding ball position. Results indicated that error in the vertical eye position was correlated with endpoint error across the conditions (Fig. S8C). This finding is consistent with the hypothesis that the eye position is modulated by the signals that encode the moment-by-moment position of the ball behind the occluder.

(3) If participants simulate accurately, then there should be no errors. But simulations cannot be perfect, so there are errors. But why would RNNs trained to simulate (but not other RNNs) be expected to make the same kind of errors as humans? Is it because some trials are harder than others when it comes to tracking latent states during the occluded epoch? It would be good to unpack the logic a little bit because it would help me better interpret this result.

We thank the reviewer for this suggestion. In this revision, we have added several new analyses that may help clarify the result.

First, prompted by the reviewers’ suggestion, we found that conditions where the vertical ball displacement during the occluded epoch is larger tended to yield larger errors for both humans and monkeys (Figure S3C). Using a process model approach, we showed that this phenomena is consistent with a noisy biased simulation with a prior towards the ball’s last visible y position (Figure S4C). In other words, the primate error patterns stem from optimizing behavior in the face of noisy simulations and in the context of an underlying prior.

We then tested whether this was true for RNNs, by estimating the correlation between their per-condition error and the vertical ball displacement during the occluded epoch, for each tested RNN model. We found that this correlation is highly predictive of the consistency scores of RNNs, with respect to both human and monkey behavior (Figure S5C). In other words, the similarity in error patterns between RNNs and primates is largely explained by this common characteristic that trials with large occluded vertical ball displacements are more difficult. We interpret this to mean that RNNs are effectively similarly biased by a similar prior, which they learn from the statistics of their training data. This is an important point to convey, which we have included in the Discussion. We thank the reviewer for prompting us to clarify it.

(4) From Figure 2A, it looks like the paddle only moved minimally during the occluded phase. Please tell me if this doesn’t make sense, but suppose you were to repeat all your analyses with paddle position taken at the start of the occluded epoch before any mental simulation could have happened, would the conclusions still hold? This might be a tall order unless you already attempted it. But could you possibly check and report the following two things?

- Correlation between errors at the start and errors at the end of the occluded epoch (something like figure 2D)
- Human-to-RNN consistency as a function of hidden-state decodability, but with consistency computed w.r.t the start of the occluded epoch (something like figure 3G).

We thank the reviewer for this suggestion. To summarize, the reviewer suggests measuring the paddle “error” using not the final paddle position, but the paddle position at the beginning of the occlusion.

We performed the analyses suggested by the reviewer and present the replication of Figure 3 when using the metric suggested here. First, the paddle position at the beginning of the occlusion is correlated with the paddle position at the end of occlusion (Figure S9A). This is expected from any observer that implements a smooth control process. Second, the human-to-RNN consistency was larger for models with simulation when we performed the comparison in the context of the paddle position at the beginning of occlusion. This result is unsurprising given the correlation between paddle position at the beginning and end of the occlusion for both humans and monkeys. As we cannot infer the motivation for this figure, we include it here as a reviewer-only figure.

A *Endpoint paddle error computed relative to occlusion start paddle position*

Point of clarification: Note that the paddle movement was not particularly small during the occluded epoch. In fact, movement per unit time was larger in the occluded epoch compared to the visible epoch (Figure S2F and S3F) for both humans and monkeys.

(5) In the same vein, could you compare the error similarity between fully visible trials and trials with occlusion for monkeys (and humans if available)? I would imagine that only the component of error that is not also present in the visible trials could be attributed to mental simulation. On a general note, I appreciate the care taken by the authors in performing all analyses on the deviation from the regression model rather than raw errors.

We thank the reviewer for this suggestion. To summarize, the reviewer suggests measuring the error with respect to the corresponding visible condition response, instead of measuring the error with respect to the ground truth ball position.

We performed the analyses suggested by the reviewer and present, in a new supplemental figure, the replication of Figure 3 when using the metric suggested here. We observe that the main inference of this work is replicated even when the pattern of errors is computed relative to the response in the visible conditions (Figure S8B).

Point of clarification: We do not equate “simulation” with “occlusion.” We instead think that the behavior in both visible and occluded conditions benefits from simulation. In the visible epoch, the computations are closed-loop: the predictions from the simulation are dynamically integrated with the stream of sensory input. In the occluded epoch, in contrast, the computations become open-loop (no sensory input) and thus the behavior more directly reflects the simulation. In other words, we view the occluder as an experimental manipulation to more precisely measure the effect of dynamic inference (simulation).

(6) The consistency in the error pattern across monkeys and humans is really striking. I am very curious to know about the extreme end point error conditions. Could you possibly show paddle positions and ground truth (like Figure 2A middle panel) for trials corresponding to the condition with most positive, most negative, and near-zero errors?

We thank the reviewer for this suggestion. We have added panels to figure 2A with example trajectories.

(7) Please correct me if I am wrong, but the test of mental simulation entails a comparison between networks with and without the ability to infer latent states during occluded epoch i.e. all_sim vs vis_sim. I think that the “no_sim” condition inflates the effect sizes in Figure 4 and 5. I say this because inference during the visual epoch requires no simulation — the participant can see the ball. So “vis_sim” is really a misnomer and should ideally be labeled “no_sim” to avoid confusion about the terminology. And “no_sim” should be labeled something else and excluded from R2 computations.

We appreciate the reviewer’s perspective but respectfully disagree with their take on when subjects make use of dynamic inference. The reviewer surmises that subjects apply two completely distinct computational strategies during the visible and occluded epochs of the task. According to their view, subjects only rely on visual input during the visual epoch, and only on dynamic inference during the occluded epoch. This viewpoint is at odds with our current understanding of control in the sensorimotor system. From a theoretical perspective, it is known that efficient control systems mitigate propagating errors by integrating sensory inputs with the output of dynamic inference engines that generate predictions. In other words, dynamic inference is a key component of control even when sensory input is available. Several lines of empirical work suggest that humans and monkeys rely on such control mechanisms. For example, the presence of predictive computations are evident in smooth pursuit where predictions help to compensate for visual latencies (Lisberger, Morris, and Tychsen 1987). The same process has been noted in manual reaching and tracking (Wolpert and Ghahramani 2004; Golub, Yu, and Chase 2015). Finally, during visual tracking, errors triggered by sudden changes of

movement direction are consistent with an underlying predictive process (Carey and Lisberger 2002). Therefore, it seems highly unlikely that dynamic inference plays no role in computations during the visual epoch. As we noted in our response to the previous comment, we don't think that the occluder triggers computations that are completely distinct from the visual epoch. Instead, we think of occlusion as a device for removing the visual feedback and thus isolating the simulation process from the other closed loop computations.

With these considerations in mind, we think that our different model types represent meaningful and important computational distinctions. The `no_sim` considers a model implements an automatized input-output function with no dynamic inference during either epoch, even though it does satisfy the task demands (minimizing endpoint error). The `vis_sim` and `all_sim` also represent meaningful differences. The `vis_sim` model only constrains the model to perform dynamic inference during the visual epoch but allows the system to create an automatized input-output function for the occluded epoch. The `all_sim`, on the other hand, constrains the models to engage in dynamic inference during both epochs.

Minor point of clarification: As we state in the "Methods - RNN Analysis": "We note that the different RNN optimization types do not correspond to mutually exclusive hypotheses, but instead map onto overlapping parts of the hypothesis space, as shown in Figure 4A. [...] For example, RNNs optimized for simulation ability only during the visible epoch ("vis-sim") could still exhibit strong simulation ability during the occluded epoch, despite not being directly optimized for this characteristic."

(8) Why is "all_sim2" worse than "all_sim" in tracking the latent state during occlude epoch? Shouldn't it be better because it has a separate readouts for ball position during occlude vs visible epochs, and consequently no interference?

To summarize, the reviewer suggests that separating the readouts should lead to better performance, given the possibility of interference. Alternatively, one could also have the opposite intuition that sharing the read-out should lead to better performance, given the similarity in the states that are read-out which should operate as a form of regularization or data augmentation. Ultimately, the difference between shared and separate read-outs may be an empirical question that depends critically on the optimization details.

(9) On a related note, what was the rationale behind using pixel-level inputs? Because ball position can be computed from images using feedforward networks, one might only need recurrence for the occluded epoch. Wouldn't it have been easier to isolate the computation of interest (and also from an RNN training standpoint) if the network was directly fed with ball position during the visual epoch? You don't need to do this, but I am curious.

We completely agree with the reviewer that modular networks that extract the latent variables from pixels in one module and perform dynamic inference in a downstream module would be a great set of candidate models to test against this behavior. This is alluded to in the Discussion where we state: "Our modeling efforts can be improved in several ways. The RNNs presented here lack many potentially relevant features of biological neural networks (e.g. spiking activity, cell types, **architectural constraints**)". This highlights a critical point of our work, that the models presented here are not the perfect models of this behavior (or the underlying neural activity), but rather a starting point from which new models can be built. We are eager to extend our work along the direction proposed but such extension requires additional assumptions that we currently are not secure in making. For example, we have to decide whether to incorporate a bottleneck in the feedforward model and how many dimensions we should allow the bottleneck to have. Similarly, we are unsure the degree to which dynamic inference computations are performed in the pixel space (e.g., in extrastriate areas with retinotopic

representations) versus in the space of latent variables (x_0, y_0, dx, dy) that are more commonly encoded in premotor areas. These limitations notwithstanding, we are excited to extend our future work in the directions noted by the reviewer.

(10) In the final analysis, isn't it odd that the ball velocity coding increases rapidly precisely when the ball is no longer visible? Wouldn't one expect it to go down because velocity information stops coming into the network? I suppose this means that the coding of velocity becomes less nonlinear and more linear as time passes. Could you please shed some light? The effect is very strong (almost 2x jump in R2), hence the question.

We thank the reviewer for bringing up this point of confusion. During the visible epoch, the RNN dynamics can be driven by the pixel input, as schematically illustrated in Figure 5B. During the occluded epoch, there is no variance in the pixel input, and so the RNN dynamics are entirely driven by recurrent computations. We interpret the rapid increase in velocity coding during the occluded epoch to mean that ball velocity is more explicit (i.e. linearly decodable) in these recurrent dynamics than in the (highly non-linear) input-driven dynamics. We have edited the text to clarify this point in the Results section.

Minor points:

(1) Figure n in the text becomes n+1 in figure captions for $n > 2$.

We thank the reviewer for finding this typo – we have fixed it in the revision.

(2) Why does the analysis in Figure S2C use absolute and not raw paddle errors?

We note that we use both absolute and signed paddle errors, to support different claims. Performance/difficulty is measured as the absolute error (MAE), as performance is typically an unsigned value, whereas patterns of errors/biases are measured via signed errors. Figure S2C uses absolute errors as a measure of condition-specific **difficulty** to support the claim that “the conditions that monkeys found to be particularly **difficult** were similarly difficult for humans.”

(3) Do results in S2D depend on the choice of parametrization (cartesian vs polar)? Are errors predicted by the distance covered after bouncing before the ball becomes invisible?

We thank the reviewer for this suggestion. In this revision, we have additionally included analysis where the initial state is parameterized in polar coordinates (Figure S4B), with no difference.

The metric of “distance-covered-after-bouncing-before-invisibility” is undefined for the majority of conditions (i.e. all conditions with no bounces or with bounces after occlusion). It is possible that the reviewer wants to know whether the “amount of visual information” provided could explain the pattern of errors. To address this potential suggestion, we additionally tested the total time during the visible epoch, and have added this to supplemental Figure S3.

References

Sohn, Hansem, and Mehrdad Jazayeri. "Validating model-based Bayesian integration using prior-cost metamers." *Proceedings of the National Academy of Sciences* 118.25 (2021).

- Carey, Megan, and Stephen Lisberger. 2002. "Embarrassed, but Not Depressed: Eye Opening Lessons for Cerebellar Learning." *Neuron* 35 (2): 223–26.
- Golub, Matthew D., Byron M. Yu, and Steven M. Chase. 2015. "Internal Models for Interpreting Neural Population Activity during Sensorimotor Control." *eLife* 4 (December). <https://doi.org/10.7554/eLife.10015>.
- Lisberger, S. G., E. J. Morris, and L. Tychsen. 1987. "Visual Motion Processing and Sensory-Motor Integration for Smooth Pursuit Eye Movements." *Annual Review of Neuroscience* 10: 97–129.
- Sohn, Hansem, and Mehrdad Jazayeri. 2021. "Validating Model-Based Bayesian Integration Using Prior-cost Metamers." *Proceedings of the National Academy of Sciences of the United States of America* 118 (25). <https://doi.org/10.1073/pnas.2021531118>.
- Wolpert, Daniel M., and Zoubin Ghahramani. 2004. "Computational Motor Control." *Science* 269: 1880–82.

Reviewers' Comments:

Reviewer #1:

Remarks to the Author:

I thought this was a good paper before the revision, and the revision itself seems like a serious, careful, and mindful addition to the original. I commend the authors for undertaking the additional steps, and recommend accepting the paper.

=====

Below are minor thoughts in response to the response-to-the-reviewers. These do **not** mean that more work is needed, or that things need to be changed or edited, and I don't expect the authors to respond to them.

1. Generalization: I agree with the authors of course that this is a serious problem in machine learning -- in scope, definition, and solution. I (of course) did not expect the authors to solve it here. I also understand why the addition of certain situations outside the 'regime' of the various models may count as far too outside their original domain, and so asking for such a test is unfair in comparison to the current standards in the field. That's all fine, and again, I'm not trying to get you to do more work or edit the paper...

BUT

Consider that your claim mainly rests on finding a (family) of simulation-based-ish RNN, and the writing makes it seem like this model should be generalizable to other simulation situations. Put it this way, **IF** what the model 'learned' is a basic simulation that allows it to go from $x,y \rightarrow x',y'$ in the domain of two-dimensional spaces that includes a single object that bounces off of hard surfaces, then...this model should handle other situations of that sort, no? Surely training it on horizontal motion and checking it on vertical motion isn't some wild generalization akin to asking you to train on the game of 'pong' and test on the game of 'asteroids'?

Or, let's put it a different way: If I had some program-learning-thing that could identifiably learn an actual simulation model for simple 2D games with rigid collisions (and such simulation models easily exist, so program-learning techniques for them exist), I would expect that program-learning-thing to be able to learn Pong and generalize it to Pong that includes a barrier in the middle of the screen.

If you think your RNN models **couldn't** handle such an addition of a barrier, it seems like either they didn't learn a simulation at all, or they learned a very weak and local kind of simulation. And what are we to make of the comparison to primates? Do they simply have a wide assortment of such weak simulations, or something more general?

Or, let's take an analogy with image recognition: If you say your model can recognize a cat, then it is reasonable on my part to expect that you will recognize a cat with a hat. It seems odd to say 'well, no, that would be outside the distribution, obviously we shouldn't expect it to do that'. Even without formally specifying what generalization is exactly, we take the English meaning of 'recognizing a cat' to include 'a cat with hat'. I take the meaning of 'our model learns a simulation of this environment' to include reasonable perturbations within the general scope of what a simulation of a two-dimensional environment with collision means.

AGAIN: It is important for me to emphasize, I do **not** expect the authors to change the revised paper, or to edit the text, do more experiments, or respond to this comment in a revision of some sort; I'm being intentionally punchy to get the point across regarding how some of the claims can read.

2. In response to point 12, the authors write: "To summarize, the reviewer is proposing the idea that (over)training drives a shift from slow, deliberate cognitive strategies to rapid, automated memorization-like strategies" --

This is super nitpicky on my part, but just to clarify: *No*, that's not what I meant.

- "Simulation" may be rapid, automated, and implicit.
- The shift may be to memorization, or to an abstraction, and these are two different things.

Again, consider the example of a ball (B) about to enter a long hallway, as follows:

=====
B -> X
=====

Will the Ball end up at point X? I'm guessing you can respond 'Yes' with or without simulating the ball. You can find signatures to differentiate these, as a non-simulation process will take you almost the same time to say 'yes' even if I make the corridor very long, in a way that does not scale with simulation. The non-simulation reasoning does not have to itself rely on memorization, BUT repeated training may drive a shift from simulation to memorization or abstraction, and all this is without committing to any of these processes being 'explicit' or 'deliberate' or 'slow'.

Reviewer #3:

Remarks to the Author:

The authors have addressed my concerns and clarified my doubts. I have no further questions.

Reviewer #1 (Remarks to the Author):

I thought this was a good paper before the revision, and the revision itself seems like a serious, careful, and mindful addition to the original. I commend the authors for undertaking the additional steps, and recommend accepting the paper.

We thank the reviewer for their thoughtful comments and questions which helped improve our work.

Below are minor thoughts in response to the response-to-the-reviewers. These do **not** mean that more work is needed, or that things need to be changed or edited, and I don't expect the authors to respond to them.

1. Generalization: I agree with the authors of course that this is a serious problem in machine learning -- in scope, definition, and solution. I (of course) did not expect the authors to solve it here. I also understand why the addition of certain situations outside the 'regime' of the various models may count as far too outside their original domain, and so asking for such a test is unfair in comparison to the current standards in the field. That's all fine, and again, I'm not trying to get you to do more work or edit the paper...

BUT

Consider that your claim mainly rests on finding a (family) of simulation-based-ish RNN, and the writing makes it seem like this model should be generalizable to other simulation situations. Put it this way, **IF** what the model 'learned' is a basic simulation that allows it to go from $x,y \rightarrow x',y'$ in the domain of two-dimensional spaces that includes a single object that bounces off of hard surfaces, then...this model should handle other situations of that sort, no? Surely training it on horizontal motion and checking it on vertical motion isn't some wild generalization akin to asking you to train on the game of 'pong' and test on the game of 'asteroids'?

Or, let's put it a different way: If I had some program-learning-thing that could identifiably learn an actual simulation model for simple 2D games with rigid collisions (and such simulation models easily exist, so program-learning techniques for them exist), I would expect that program-learning-thing to be able to learn Pong and generalize it to Pong that includes a barrier in the middle of the screen.

If you think your RNN models **couldn't** handle such an addition of a barrier, it seems like either they didn't learn a simulation at all, or they learned a very weak and local kind of simulation. And what are we to make of the comparison to primates? Do they simply have a wide assortment of such weak simulations, or something more general?

Or, let's take an analogy with image recognition: If you say your model can recognize a cat, then it is reasonable on my part to expect that you will recognize a cat with a hat. It seems odd to say 'well, no, that would be outside the distribution, obviously we shouldn't expect it to do that'. Even without formally specifying what generalization is exactly, we take the English meaning of 'recognizing a cat' to include 'a cat with hat'. I take the meaning of 'our model learns a simulation of this environment' to include reasonable perturbations within the general scope of what a simulation of a two-dimensional environment with collision means.

AGAIN: It is important for me to emphasize, I do **not** expect the authors to change the revised paper, or to edit the text, do more experiments, or respond to this comment in a revision of some sort; I'm being intentionally punchy to get the point across regarding how some of the claims can read.

We thank the reviewer for taking the time to expand on this thought. We largely share the intuition that the generalization abilities of today's models in our field (and beyond) do not align with our own generalization abilities.

2. In response to point 12, the authors write: "To summarize, the reviewer is proposing the idea that (over)training drives a shift from slow, deliberate cognitive strategies to rapid, automated memorization-like strategies" --

This is super nitpicky on my part, but just to clarify: *No*, that's not what I meant.

- "Simulation" may be rapid, automated, and implicit.
- The shift may be to memorization, or to an abstraction, and these are two different things.

Again, consider the example of a ball (B) about to enter a long hallway, as follows:

```
=====
B -> X
=====
```

Will the Ball end up at point X? I'm guessing you can respond 'Yes' with or without simulating the ball. You can find signatures to differentiate these, as a non-simulation process will take you almost the same time to say 'yes' even if I make the corridor very long, in a way that does not scale with simulation. The non-simulation reasoning does not have to itself rely on memorization, BUT repeated training may drive a shift from simulation to memorization or abstraction, and all this is without committing to any of these processes being 'explicit' or 'deliberate' or 'slow'.

We thank the reviewer for clarifying this point. Indeed, it is possible/likely that simulations are not synchronous with external object dynamics, nor explicit! We believe it would be useful, moving forward, for the field to agree on terminology to describe the large space of solutions.

Reviewer #3 (Remarks to the Author):

The authors have addressed my concerns and clarified my doubts. I have no further questions.

We thank the reviewer for their thoughtful comments and questions which helped improve our work.